# Hierarchical Scoring with 3D Gaussian Splatting for Instance Image-Goal Navigation

## Abstract

Instance Image-Goal Navigation (IIN) requires autonomous agents to identify and navigate to a target object or location depicted in a reference image captured from any viewpoint. While recent methods leverage powerful novel view synthesis (NVS) techniques, such as 3D Gaussian splatting (3DGS), they typically rely on randomly sampling multiple viewpoints or trajectories to ensure comprehensive coverage of discriminative visual cues. This approach, however, creates significant redundancy through overlapping image samples and lacks principled view selection, substantially increasing both rendering and comparison overhead. In this paper, we introduce a novel IIN framework with a hierarchical scoring paradigm called **GauScoreMap** that estimates optimal viewpoints for target matching. Our approach integrates cross-level semantic scoring, utilizing CLIP-derived relevancy fields to identify regions with high semantic similarity to the target object class, with fine-grained local geometric scoring that performs precise pose estimation within promising regions. Extensive evaluations demonstrate that our method achieves state-of-the-art performance on simulated IIN benchmarks and real-world applicability.

## 1 Introduction

Instance Image-Goal Navigation (IIN) is critical in embodied navigation, requiring an agent to identify and move to the object or location depicted in a target image—often captured from any viewpoint Krantz et al. (2022). This flexibility is essential in real-world scenarios where users may provide photos from arbitrary perspectives. However, viewpoint discrepancies, cluttered scenes, and occlusions complicate the alignment of target images with the agent's observations. Effective solutions must therefore reason not only about what object appears in the image, but also where and from which viewpoint the image could have been taken inside the 3D environment.

Motivated by advances in novel view synthesis (NVS) methods, such as Neural Radiance Fields (NeRF) Mildenhall et al. (2021) and 3D Gaussian splatting (3DGS) Kerbl et al. (2023), recent approaches have begun to explore more expressive, view-consistent scene representations for IIN. Methods Cui et al. (2024); Wang et al. (2024) combine NeRF rendering with a topological graph, embedding RGB observations and learned image features into graph nodes. While this strategy preserves appearance information, discretizing the environment into nodes constrains the agent's ability to observe the scene from arbitrary views and limits truly free-view navigation.

Alternatively, 3DGS-based approaches Lei et al. (2025); Meng et al. (2024); Honda et al. (2025) maintain a continuous 3D representation with high rendering fidelity. Yet these methods typically resort to pose-centric search strategies by sampling many viewpoints or trajectories to ensure coverage of discriminative image cues. Such brute-force sampling is inefficient in continuous 6-DoF space, where even dense sampling may miss informative viewpoints and where redundant rendering substantially increases computation. The core challenge lies not only in how to render more efficiently, but in how to reason about where the target image is likely to have been generated within the 3D scene.

To address this, we propose **GauScoreMap**, a new IIN framework that shifts the perspective from pose-centric sampling to field-centric inference. Instead of treating viewpoint selection as an unstructured search problem, we reinterpret the 3DGS scene as a *joint semantic–geometric field*: each Gaussian carries both appearance descriptors and explicit 3D geometry, making the scene amenable

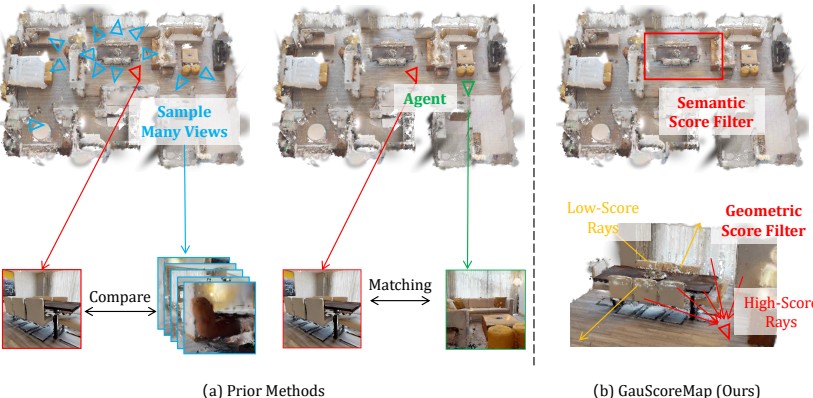

(a) Prior Methods      (b) GauScoreMap (Ours)

Figure 1: Overall method comparison. Prior approaches typically (i) sample many views around candidate objects or (ii) struggle to match visually dissimilar images. Our method leverages a 2-stage scoring method to semantically and geometrically locate the target image efficiently.

to dense, continuous scoring. Building upon this view, our method performs hierarchical scoring over the 3DGS representation to infer where the target image content most plausibly originates.

First, our global semantic scoring constructs a soft semantic relevance field by computing cosine similarity between CLIP text embeddings extracted from the target image and per-Gaussian semantic features. This yields coherent candidate regions that reflect class-level likelihood across the entire 3D scene. Second, our local geometric scoring evaluates each region at the ray level: by comparing sampled Gaussian-ray features with DINOv2-derived image features via cross-attention, we estimate a ray-wise camera likelihood that directly reflects how well rays in the scene explain the target image. This transforms the fine-level matching problem from "render many views and compare them" into the more principled objective of "infer where in the 3D semantic–geometric field the target rays most likely emanated," enabling accurate pose recovery without exhaustive rendering.

Recent advances in 3DGS have enabled embodied agents to reconstruct and explore environments with high visual fidelity, but existing methods still struggle when appearance varies significantly between the target image and agent observations Lei et al. (2025); Meng et al. (2024) (Figure 1). Our formulation addresses this gap by explicitly integrating global semantic cues with local geometric reasoning in a unified scoring field. This yields robust, viewpoint-invariant localization while dramatically reducing computation, enabling generalization across diverse scenes and target viewpoints.

Our contributions are summarized as follows: 1) We introduce a hierarchical scoring formulation that views 3DGS as a joint semantic–geometric field, enabling principled inference over where the target image content most likely appears in 3D space. 2) We leverage this score field to identify and prioritize the most informative viewpoints for matching, eliminating the need for exhaustive viewpoint sampling. 3) Our method achieves new state-of-the-art results on instance-specific image-goal navigation benchmarks and demonstrates strong robustness and efficiency in real-world indoor environments.

## 2 RELATED WORK

### 2.1 INSTANCE IMAGE GOAL NAVIGATION

Deep reinforcement learning approaches have emerged as a major solution to IIN, where end-to-end policies are learnt to align current observations with target images, achieving promising simulator results through extensive training Lei et al. (2024); Qin et al. (2025). However, these reactive methods struggle to retain knowledge of explored areas in complex scenes Krantz et al. (2023), lacking explicit environment representations and degrading when agents must re-localize after losing sight of key features. To improve context retention and adaptability, map-based IIN methods incorporate spatial representations to guide navigation Yu et al. (2023); Majumdar et al. (2022); Yuan et al.

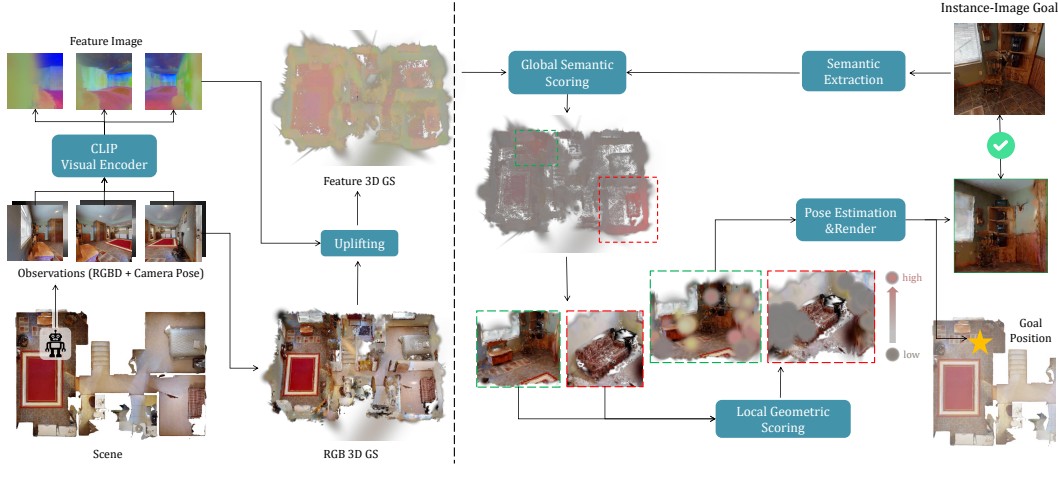

Figure 2: Overview of our GauScoreMap approach for Instance Image-goal Navigation. Our method consists of two main stages: (a) Gaussian Reconstruction and Feature Lifting, where we build a 3D Gaussian representation of the environment and lift CLIP features into this representation; and (b) Instance-Image Goal Localization, which uses a two-step scoring process to first identify semantically relevant regions and then precisely locate the target object instance.

(2024). Early approaches used metric maps—typically 2D bird's-eye view grids from SLAM—to track spatial locations relative to obstacles and landmarks Chaplot et al. (2020); Lei et al. (2024). While addressing some reactive strategy limitations, these 2D representations discard 3D geometry and texture details crucial for goal image matching. Recent work explores structured maps and novel view synthesis Cui et al. (2024); Wang et al. (2024); Lei et al. (2025); Meng et al. (2024); Honda et al. (2025), combining visual features with graph nodes to preserve rich appearance cues. These developments highlight that robust IIN requires representations bridging environment structure with high-resolution visual information for accurate target localization.

## 2.2 NOVEL VIEW SYNTHESIS IN EMBODIED VISUAL NAVIGATION

Early efforts like e2e-NeRF-nav Liu et al. (2024b) integrate online Neural Radiance Fields into the control loop for end-to-end training, but continual NeRF Mildenhall et al. (2021) updates are computationally expensive. HNR-VLN Wang et al. (2024) shifts complexity from policy learning to look-ahead synthesis by using NeRF to render candidate viewpoints for graph search. Frontier-enhanced Topological Memory Cui et al. (2024) extends this by adding "ghost" nodes to topological graphs, combining geometric reachability with appearance-based reasoning. However, discrete node representations limit diverse viewpoint observation and hinder navigation in complex layouts. GaussNav Lei et al. (2025) instead uses 3D Gaussian Splatting Kerbl et al. (2023) to preserve high-fidelity geometry and textures, while BEINGS Meng et al. (2024) employs Monte Carlo model-predictive control with hypothetical rollout rendering. Despite their effectiveness, 3DGS-based methods suffer from high computational overhead, motivating our development of a method that leverages fine-grained local visual information without extensive trajectory or viewpoint sampling.

## 3 METHOD

### 3.1 OVERVIEW

In Instance Image-goal Navigation (IIN), an agent navigates to a specific object instance shown in a goal image $I_g$. Starting from an initial position and orientation, the agent receives RGB-D observations and camera poses at each timestep, selecting actions to locate the target. Success is achieved when the agent reaches the goal vicinity within a maximum action limit.

To address IIN, we propose *Gaussian Splatting Score Maps for Visual Navigation* (GauScoreMap), illustrated in Figure 2. Our method operates in two stages: First, the agent explores the environment to build a Gaussian splatting representation and lifts 2D visual features into a 3D feature-rich Gaussian field. Second, we perform hierarchical scoring—extracting semantic information from the goal image to generate a global score map identifying candidate regions, then computing local similarity scores within these regions for precise target localization.

## 3.2 GAUSSIAN RECONSTRUCTION AND FEATURE LIFTING

### 3.2.1 GAUSSIAN RECONSTRUCTION

When placed in a new environment, the agent employs a frontier-based exploration strategy Yamauchi (1997); Holz et al. (2010); Juliá et al. (2012) to systematically cover the environment and collect observations for Gaussian reconstruction.

From the collected observations $\{(I_i, D_i, P_i) | i \in [0, N]\}$ (RGB images, depth maps, and camera poses), we reconstruct the RGB Gaussian splatting field using a hierarchical approach Yugay et al. (2023; 2024). For each observation subset $\{(I_i, D_i, P_i) | i \in [m, n]\}$, we initialize a submap by backprojecting the first frame's RGB image into 3D space using depth and pose. Subsequent frames densify the submap with additional Gaussian primitives. Finally, local submaps are merged into a global field.

During training, we render both color and depth from the Gaussian field. For pixel $p$ viewing from direction $v$, the color and depth values are computed from the ordered set $\mathcal{S}_{v,p}$ of Gaussis:

$$\hat{I}_v(p) = \sum_{i \in \mathcal{S}_{v,p}} c_i(v) w_i(v, p), \quad \hat{D}_v(p) = \sum_{i \in \mathcal{S}_{v,p}} d_i(v) w_i(v, p) \tag{1}$$

where $\hat{I}_i(p)$ and $\hat{D}_i(p)$ are rendered color and depth values, $c_i(v)$ and $d_i(v)$ are color and depth values of the $i$-th Gaussian along the ray through pixel $p$ viewing from direction $v$, $w_i(v, p) = \alpha_i(v, p) \prod_{j \in \mathcal{S}_{v,p}, j < i}(1 - \alpha_j(v, p))$ is the rendering weight and $\alpha_i(v, p)$ is the transparency value.

The optimization uses a combined loss as $\mathcal{L} = \lambda_{\text{color}} \mathcal{L}_{\text{color}} + \lambda_{\text{depth}}$, where $\mathcal{L}_{\text{color}} = \|I_i - \hat{I}_i\|_1$ and $\mathcal{L}_{\text{depth}} = \|D_i - \hat{D}_i\|_1$ are L1 losses between ground truth and rendered images/depths, and $\lambda$ terms balance the loss components.

### 3.2.2 FEATURE LIFTING

The visual features produced by the CLIP Radford et al. (2021) visual encoder are uplifted with simple aggregation from all collected frames Marrie et al. (2024). For each 3D Gaussian in the scene, we construct its feature representation as a weighted average of 2D features from all frames. The feature $f_i$ of Gaussian $i$ is:

$$f_i = \sum_{(v,p) \in S_i} \bar{w}_i(v, p) F_{v,p} \text{ with } \bar{w}_i(v, p) = \frac{w_i(v, p)}{\sum_{(v,p) \in S_i} w_i(v, p)} \tag{2}$$

where $S_i = \{(v, p) : i \in \mathcal{S}_{v,p}\}$ is the set of view-pixel pairs passing through Gaussian $i$. This weighting is intuitive: larger rendering weights indicate closer proximity to ray termination, so corresponding features $F_{v,p}$ contribute more significantly to Gaussian $i$'s representation.

## 3.3 GLOBAL SEMANTIC SCORING

After constructing the CLIP feature Gaussian field, we generate a global relevancy score map from the visual input. Since images contain more than just the target object, directly computing relevancy with the feature field produces noisy segmentation. We therefore use Mask-RCNN He et al. (2017) to extract the class label, which is fed to the CLIP text encoder to obtain text embedding $E_T$. For each Gaussian $g$ with CLIP feature $f_g$, we compute a relevancy score $S_g$ using cosine similarity:

$$S_g = \frac{E_T \cdot f_g}{\|E_T\| \cdot \|f_g\|} \tag{3}$$

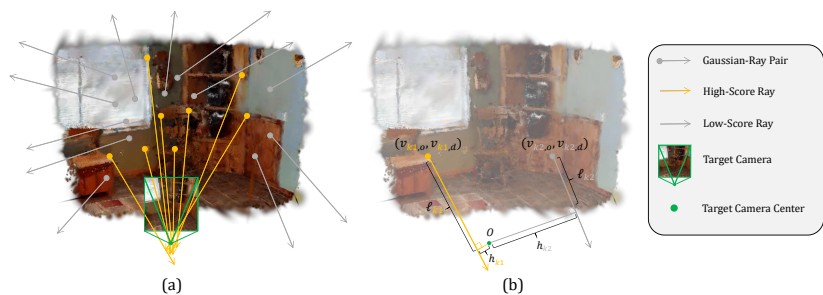

(a)  (b)

Figure 3: An example of Local Geometric Scoring. (a) High-Score Rays cluster tightly around the ground-truth camera center, whereas Low-Score Rays diverge and spread outward. (b) Illustration showing that the high-score ray k1 yield smaller values of $h$ than the low-score ray k2, defined as the distance between the ground-truth camera center and its orthogonal projection onto the ray.

This assigns a relevancy score to each Gaussian, creating a continuous score field. Applying a threshold $\tau$ yields segmented regions of Gaussians likely belonging to the target category. To address fragmentation from naive thresholding, we follow LUDVIG Marrie et al. (2024) by incorporating scene geometry and diffusing the segmentation based on feature similarity between neighboring Gaussians. This connects fragmented parts of the same instance into well-formed connected components, each representing a candidate region containing a potential target. These regions significantly reduce the search space for subsequent image-based 6D pose estimation, where we match the goal image against each candidate to precisely locate the target instance.

## 3.4 LOCAL GEOMETRIC SCORING

After identifying multiple candidate regions through global semantic scoring, we need to further refine our search in two stages: first determining the most likely local region containing the target object, and then estimating the precise 6D pose within that region.

### 3.4.1 LOCAL SCORING FOR REGION SELECTION

For each candidate region identified in the global scoring stage, we sample random Gaussians and generate rays in the hemisphere defined by the surface normal of each Gaussian (estimated using neighboring Gaussians). This process yields a set of ray inputs: $\{(o_i, d_i, c_i)|i \in [0, K]\}$, where $o_i$ is the ray origin, $d_i$ is the ray direction, and $c_i$ is the 1-st order spherical harmonic coefficient representing the color of the ray.

Following the approach in Matteo et al. (2024), we encode these rays using a learned MLP with positional encoding as $r_i = \text{MLP}(\gamma(o_i), \gamma(d_i), \gamma(c_i))$, where $\gamma(\cdot)$ denotes the positional encoding function, transforming the ray set into a feature representation of shape $(K, C_1)$.

Concurrently, we process the goal image $I_g$ through a DINOv2 Oquab et al. (2023) visual encoder to get its visual feature $F_g$ of shape $(L, C_2)$, where $L = h \times w$ represents the spatial dimensions of the feature map. These features are then compared through a cross-attention mechanism:

$$A = \text{CrossAttention}(r, F_g) \in \mathbb{R}^{K \times L} \tag{4}$$

To obtain a single relevance score for each ray, we sum the cross-attention values along the second dimension, giving $\hat{s}_k = \sum_{l=1}^{L} \hat{A}_{k,l}$. During training, both the ray MLP and the cross-attention module are supervised to match these predicted scores $\hat{s}_k$ with geometric ground-truth scores $s_k$.

The key idea behind computing the ground-truth score $s_k$ is that relevant rays should ideally converge toward the ground-truth camera center of the target image. If a ray has high relevance score, the camera center should lie directly on or very close to that ray. Therefore, we use the perpendicular distance between the ray and the true camera center as a geometric indicator of relevance. And we show an example of calculating the perpendicular distance in Figure 3.

Formally, for each sampled ray $k$, we compute the distance $h_k = \|(v_{k,o} + \ell_k v_{k,d}) - \mathbf{O}\|_2$, where $v_{k,o}$ and $v_{k,d}$ denote the ray origin and unit direction, $\mathbf{O}$ is the ground-truth camera center, and $\ell_k = \max\left((\mathbf{O} - v_{k,o}) \cdot v_{k,d}, \, 0\right)$ is the projection of the vector $(\mathbf{O} - v_{k,o})$ onto the ray, truncated to ensure it lies in front of the ray origin. This projection finds the closest point on the ray to the camera center. The resulting distance $h_k$ serves as the geometric ground-truth score: rays that pass close to the true camera center have small $h_k$, whereas irrelevant rays yield large distances. Values range from $0$ (the ray exactly intersects the camera center) to arbitrarily large numbers.

Finally, we map the distance $h_k$ of the $k$-th ray to its ray score using:

$$\delta_k = 1 - tanh(h_k), \quad s_k = \delta_k \frac{L}{\sum_{i=1}^{K} \delta_i} \tag{5}$$

We apply a $\tanh(\cdot)$ mapping because $h_k \in [0, +\infty)$ and $\tanh(\cdot)$ smoothly compresses this range into $[0, 1)$. And the $\delta_k = 1 - tanh(h_k)$ maps $[0, 1)$ to $[1, 0)$, which provides a geometric confidence: smaller distances $h_k$ produce values of $\delta_k$ closer to $1$, but larger distances push $\delta_k$ toward $0$.

To obtain an initial geometric ray score, we first normalize the values across all sampled rays as $\frac{\delta_k}{\sum_{i=1}^{K} \delta_i}$. To match this geometric score with the scale of the predicted score, we multiply the normalized term by $L$. This is because a softmax is applied across the $K$ rays for each of the $L$ feature locations in the attention map $A$, ensuring that the attention weights $\{\hat{A}_{1,l}, \hat{A}_{2,l}, \ldots, \hat{A}_{K,l}\}$ sum to $1$ for every feature index $l$. Since the predicted ray score $\hat{s}_k$ is computed by summing these attention weights over the $L$ feature dimensions, its magnitude naturally scales with $L$.

Then we train the MLP and cross attention modules by minimizing the $L2$ loss $\mathcal{L} = \frac{1}{K} \sum_{k=1}^{K} ||s_k - \hat{s}_k||$ between the predicted ray scores $\hat{S}$ and the ground truth ray scores $S$.

### 3.4.2 Fine-Grained Pose Estimation for Precise Localization

Once we've identified the most promising region, we perform a second, more dense sampling of Gaussian-ray pairs within this region. This denser sampling allows for more precise localization of the target object. We select top $k$ Gaussian-ray pairs with the highest scores from the new samples, and perform triangulation as described in Matteo et al. (2024) to estimate the 6D pose (position and orientation) of the target object. This two-stage scoring approach—first at the region level and then at the pose level—enables our system to efficiently narrow down the search space before performing precise localization, significantly improving both the efficiency and accuracy of the object localization process. These two scoring steps use the same pretrained ray-image cross attention neural network by minimizing the difference between the predicted camera 6D pose and the gt camera 6D pose as Matteo et al. (2024).

## 4 Experiment

### 4.1 Experiment Setup

**Dataset.** We conduct our experiments using the Habitat Szot et al. (2021) simulator. For scene data, we utilize the Habitat-Matterport 3D dataset (HM3D) Yadav et al. (2023). Specifically, we use version 0.2 of the HM3D dataset and follow the Instance ImageGoal Navigation (IIN) Krantz et al. (2022) in the Habitat Navigation Challenge 2023[1]. We evaluate our method on the 1,000 validation episodes specified by Krantz *et al.* Krantz et al. (2022). This validation subset encompasses six object categories: {*chair, couch, bed, toilet, television, plant*} and includes 795 unique object instances.

**Agent Configuration.** We adopt the standard agent configuration from the Habitat Navigation Challenge 2023. The agent is modeled as a rigid-body cylinder with zero turning radius, standing 1.41m tall with a radius of 0.17m. A forward-facing RGB-D camera is mounted at a height of 1.31m. At each time step $t$, the agent receives observations consisting of RGB images, depth maps, and sensor poses. The agent operates in a continuous action space with four dimensions: *linear velocity,*

---

[1]https://aihabitat.org/challenge/2023/

Table 1: Success rate and SPL comparison of our method with four sets of baseline methods.

| Category | Method | SR ↑ | SPL ↑ |
|---|---|---|---|
| MultiON Transfer | MultiON Baseline Wani et al. (2020) | 0.066 | 0.045 |
| | MultiON Implicit Marza et al. (2023) | 0.143 | 0.107 |
| | MultiON Camera Chen et al. (2022) | 0.186 | 0.142 |
| SOTA IIN | Mod-IIN Krantz et al. (2023) | 0.561 | 0.233 |
| | IEVE Mask RCNN Lei et al. (2024) | 0.684 | 0.241 |
| | IEVE InternImage Lei et al. (2024) | 0.702 | 0.252 |
| SOTA IIN with Scene/3DGS Map | Mod-IIN (Scene Map) Krantz et al. (2023) | 0.563 | 0.323 |
| | IEVE Mask RCNN (Scene Map) Lei et al. (2024) | 0.683 | 0.331 |
| | IEVE InternImage (Scene Map) Lei et al. (2024) | 0.705 | 0.347 |
| | GaussNav (3DGS Map) Lei et al. (2025) | 0.725 | 0.578 |
| | GauScoreMap (3DGS Map) | **0.784** | **0.605** |

*angular velocity, camera pitch velocity, and velocity stop.* Each action dimension accepts values between -1 and 1, which are then scaled according to their respective configuration parameters. The maximum linear speed is $35cm/frame$, while the maximum angular velocity is $60°/frame$.

**Evaluation Metrics.** Our evaluation incorporates both effectiveness and navigation efficiency metrics. The primary metrics we use are SR (Success Rate) and SPL (Success weighted by Path Length). A navigation attempt is considered successful when the agent executes the stop action within a 1.0m radius of the target object and can visually detect the object by adjusting its camera orientation. The SPL metric, as introduced by Anderson et al. Anderson et al. (2018), provides a balanced assessment of navigation efficiency by considering both success and path optimality.

## 4.2 COMPARISON WITH STATE-OF-THE-ART METHODS

We compare our method against a comprehensive set of baseline approaches as presented in Table 1, with baseline results sourced from GaussNav Lei et al. (2025). The comparison methods are organized into three categories:

**MultiON Transfer Methods.** These approaches were originally designed for the MultiON task, which shares similarities with scene-specific map representations: (1) MultiON Baseline, a standard implementation of Wani et al. (2020); (2) MultiON Implicit Marza et al. (2023), which learns an implicit neural representation; and (3) MultiON Camera Chen et al. (2022), which develops an active camera movement policy.

**State-of-the-art IIN Methods.** Leading IIN approaches include: (1) Mod-IIN Krantz et al. (2022), which decomposes the task into exploration, goal instance re-identification, goal localization, and local navigation; (2) IEVE Mask RCNN Lei et al. (2024), which implements a modular architecture using Mask RCNN He et al. (2017) for object detection; and (3) IEVE InternImage, an enhanced variant with a more powerful detector.

**SOTA IIN with Scene/3DGS Map.** This category includes the above methods when augmented with different map representations: (1-3) Mod-IIN, IEVE Mask RCNN, and IEVE InternImage with traditional scene maps; (4) GaussNav Lei et al. (2025), which utilizes 3D Gaussian Splatting maps; and (5) our proposed approach, which also leverages 3DGS maps but enhances performance.

As shown in Table 1, our method significantly outperforms all baselines, achieving the highest success rate (0.784) and SPL (0.605). MultiOn methods perform poorly because they were designed for object-goal tasks, not instance-goal tasks. Their failures often stem from finding the correct object category but not the specific instance. Notably, our approach surpasses GaussNav by 5.9% in success rate and 2.7% in SPL. While both GaussNav Lei et al. (2025) and our method utilize Gaussian splatting fields for navigation, our method has better performance with our enhanced localization capabilities.

Table 2: Ablation study of our method.

| Method | SR ↑ | SPL ↑ |
|---|---|---|
| GauScoreMap | 0.784 | 0.605 |
| GauScoreMap w.o. Global Semantic Scoring | 0.608 | 0.419 |
| GauScoreMap w.o. Local Geometric Scoring | 0.421 | 0.310 |
| GauScoreMap w. GT Match | 0.842 | 0.650 |
| GauScoreMap w. GT Global Localization | 0.944 | 0.742 |

## 4.3 ABLATION STUDY

We ablate the main design choices of our method and show their influences on the final performance in Table 2. And we analyze each module of our method:

**GauScoreMap w.o. Global Semantic Scoring.** The global semantic scoring module serves as a prefilter to extract candidate local regions for finer local localization. From Table 2, we can see that without global semantic scoring, with only local geometric scoring to produce the predicted target position, the success rate drops by 17.6%. This is because indoor scenes have severe occlusion and complicated spatial distributions. Simply sampling Gaussians and computing the relationships between ray features and Gaussians suffers from ambiguity and inefficiency. The global semantic scoring effectively narrows down the search space by identifying semantically relevant regions first, allowing the local geometric scoring to focus on promising areas. This two-stage approach significantly improves both accuracy and computational efficiency compared to relying solely on local visual scoring.

**GauScoreMap w.o. Local Geometric Scoring.** When we remove the local geometric scoring component and rely only on global semantic scoring, performance decreases dramatically with a 36.3% drop in success rate. This substantial decline highlights the critical role of fine-grained geometric matching in precisely localizing the target object. While global semantic scoring can identify candidate regions containing objects of the target category, it lacks the precision to distinguish specific object instances with similar semantic properties. The local geometric scoring module provides this crucial instance-level discrimination by establishing detailed geometric correspondences between the goal image features and the 3D scene representation.

**GauScoreMap w. GT Match.** Using ground truth instance matching improves success rate by 5.8%, revealing room for enhancement in instance-level recognition and matching. This gap suggests that better visual feature extraction and matching could further boost performance.

**GauScoreMap w. GT Global Localization.** With ground truth global localization, our method achieves 94.4% success, demonstrating highly effective local navigation from accurate position estimates. This indicates that global localization is the primary bottleneck, and improving the global semantic scoring module could approach this upper bound.

## 4.4 TIME EFFICIENCY AND PEAK VRAM USAGE EVALUATION

Table 3 evaluates the time efficiency and peak VRAM usage of our method compared with Gauss-NavLei et al. (2025). We analyze each substage across training and inference phases using three HM3D Yadav et al. (2023) scenes averaging 80 $m^2$ with ∼1000 images each. In the training stage, compared to GaussNav Lei et al. (2025), our submap division strategy reduces GS reconstruction time by 50 minutes while requiring only one-third of the VRAM. For local scoring function training, we save 30 minutes over 6DGS Matteo et al. (2024) (45 minutes) by restricting the camera pose search space to navigable areas defined by the reconstructed scene. Our total inference time (2.3s) is way faster than that of GaussNavLei et al. (2025) (30s-50s), this is attributed to our efficient local scoring strategy, rather than render several full views for comparison in GaussNav, we only sample a small batch of rays and conduct the ray-image compare to locate the targe image. Among all our inference substages, all of them are efficient except global scoring, which computes relevancy for all Gaussians in the scene. We address this by sparsifying the GS scene to ∼600,000 Gaussians, and this preserves semantic accuracy since CLIP features remain embedded in target objects regardless of Gaussian count due to their distinctive semantic properties.

Table 3: Time Efficiency and Peak VRAM usage of our method compared with GaussNav Lei et al. (2025). (The metrics of GaussNav are denoted using gray color.)

| Stage | Substage | Time | Peak VRAM |
|---|---|---|---|
| GaussNav Training | Gaussian Reconstruction | 65 m | 11.0 GB |
| Ours Training | Gaussian Reconstruction | 15 m | 3.6 GB |
| | Local Scoring Function Training | 15 m | 5.5 GB |
| GaussNav Inference | Semantic Localization | 0.10 s | 3.5 GB |
| | NVS for Local Localization | 30 s-50 s | 4.0 GB |
| Ours Inference | Semantic Extraction (Mask-RCNN forward) | 0.10 s | 3.5 GB |
| | Global Scoring (Relevancy score calculation) | 1.12 s | 1.1 GB |
| | Local Scoring (Gaussian Sampling) | 0.59 s | - |
| | Local Scoring (Candidate Selection) | 0.32 s | 4.2 GB |
| | Local Scoring (Pose Estimation) | 0.17 s | 4.2 GB |

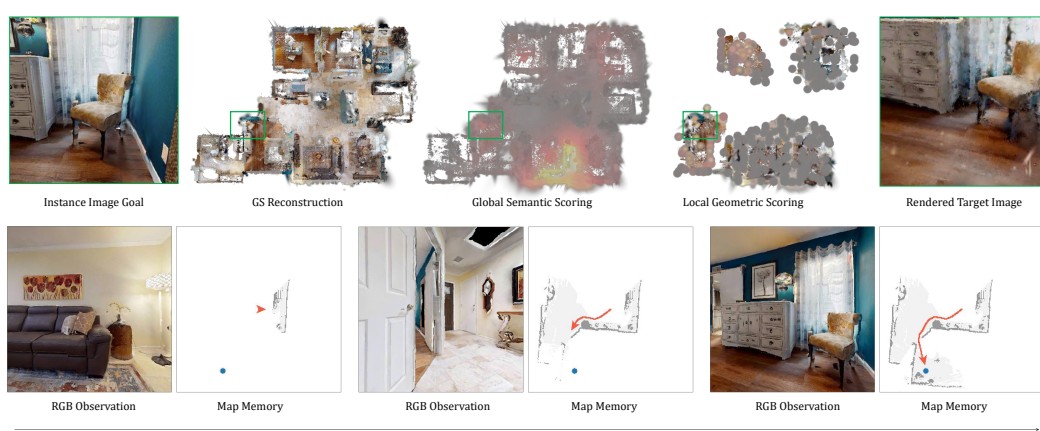

Figure 4: The localization by scoring (the first row) and the navigation process (the second row) of an episode of scene *5cdEh9F2hJL* in HM3D Yadav et al. (2023).

### 4.5 VISUAL SCORING AND NAVIGATION RESULTS

Figure 4 illustrates the scoring and navigation results of our method. The first row demonstrates how our hierarchical scoring approach localizes the target image and renders it using reconstructed Gaussian splats. The second row shows the agent navigating to the identified position in a Habitat simulator Szot et al. (2021). Additional examples are provided in the appendix.

## 5 CONCLUSION

In this work, we introduce a novel Instance Image-Goal Navigation framework that tackles the principal challenges of viewpoint variation, semantic ambiguity, and complex scene layouts. By combining two-level semantic scoring with fine-grained geometric scoring, the method yields a continuous score map that obviates the need for exhaustive or random viewpoint sampling. Empirical evaluations on simulated benchmarks confirm state-of-the-art performance, underscoring the method's effectiveness and practical applicability. Furthermore, we deploy the proposed approach on a humanoid agent and validate its performance in real-world indoor environments. A key limitation of our method is that it focuses primarily on static environments and relies on a pre-built GS scene map. Future directions may include simultaneously exploring and locating the target image without accuracy loss.

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

# A APPENDIX

## A.1 EXTRA SIMULATION RESULTS

We present two additional examples of our method navigating to instance image goals in Figure 5 and Figure 6. Each example demonstrates both the target localization process (first row) and the navigation execution (second row).

In the first row, we illustrate the complete target localization pipeline from left to right: the instance image goal, the Gaussian Splatting (GS) reconstruction of the scene, the heat map generated during the global semantic scoring step, the local score map within segmented candidate regions, and the rendered target image using the pose estimated by the local geometric scoring step.

The second row shows three key stages of the navigation process as the agent moves toward the final goal position.

## A.2 LOCALIZATION OF OTHER OBJECTS

Since our method encodes CLIP features into the Gaussian field, it can localize objects beyond the six evaluation categories (*chair, couch, bed, toilet, television, plant*) in HM3D Yadav et al. (2023). Figure 7 demonstrates successful localization of *lamp*, *refrigerator*, and *bicycle* in Gibson Xia et al. (2018) and ReplicaCAD Straub et al. (2019) scenes.

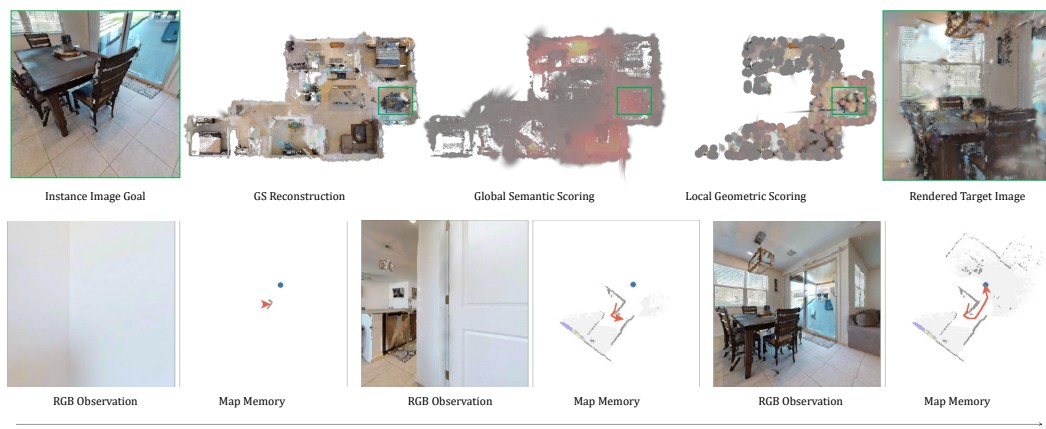

Figure 5: The target localization (the first row) and the navigation process (the second row) of an episode of scene *Nfvxx8J5NCo* in HM3D Yadav et al. (2023)'s validation set.

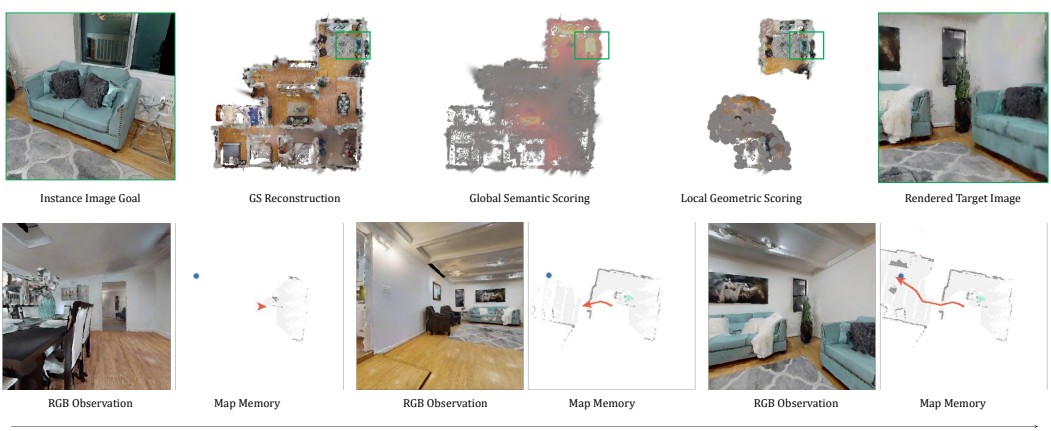

Figure 6: The target localization (the first row) and the navigation process (the second row) of an episode of scene *Dd4bFSTQ8gi* in HM3D Yadav et al. (2023)'s validation set.

### A.3 COMPARISON WITH OTHER NERF/GS-BASED LOCALIZATION METHODS

Since localization is the key of our method for successful IIN navigation, we validate our localization module by replacing only this component while keeping other parts unchanged and compare against alternative methods. For NeRF-based approaches, we combine iNeRF Yen-Chen et al. (2021) for pose estimation with NeRF-SLAM Rosinol et al. (2023) for scene reconstruction. For 3DGS methods, we evaluate 6DGS Matteo et al. (2024) and GS-CPR Liu et al. (2024a), both operating on our reconstructed 3DGS maps.

We evaluate on a 100-episode subset of the HM3D validation set, with navigation success rates shown in Table 4.

iNeRF achieves 0% SR due to NeRF-SLAM's reconstruction failures: the system exhausts GPU memory when allocating large implicit volumes for HM3D apartments. This reflects well-documented limitations of implicit NeRF pipelines in large, complex indoor scenes Tancik et al. (2022); Turki et al. (2022). Gaussian splatting scales better through decomposition into smaller, independent segments.

GS-CPR also achieves 0% SR because it requires close spatial overlap between query and database images—rarely satisfied in large-scale scenarios. 6DGS attains 58% SR, respectable given the localization difficulty in complex indoor structures, but substantially below our method's performance.

| Instance Image Goal | 3DGS Reconstruction | Global Semantic Scoring | Local Geometric Scoring | Zoom-in Candidate 3DGS | Rendered Target Image |
|---|---|---|---|---|---|

Figure 7: Localization of *Lamp* (first row), *Refridgerator* (second row) and *Bicycle* (third row) on Gibson and ReplicaCAD scenes.

| Method | SR |
|---|---|
| iNeRF Yen-Chen et al. (2021) | 0 (OOM) |
| GS-CPR Liu et al. (2024a) | 0 |
| 6DGS Matteo et al. (2024) | 0.58 |
| GauScoreMap (Ours) | 0.76 |

Table 4: Navigation SR comparison with different localization methods

These results demonstrate that our method achieves an effective balance between computational efficiency and navigation performance in the IIN task, substantially outperforming alternative localization approaches in large-scale indoor environments.

## A.4 REAL WORLD EXPERIMENTS

### A.4.1 ENVIRONMENT

The layout of the testing field used for the demonstrations is illustrated in Figure 8. The environment is intentionally cluttered with numerous common household items to closely mimic real-world conditions. Additionally, several boxes are strategically placed throughout the area to introduce obstacles, thereby increasing the complexity and challenging the robot's navigation and detection capabilities.

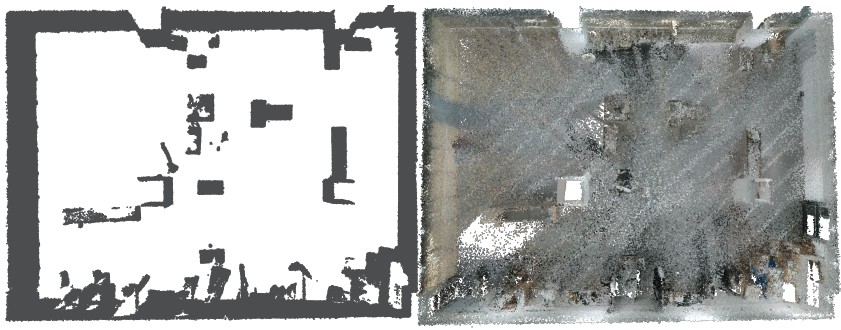

Figure 8: Layout of the testing field (left) along with point-cloud construction (right).

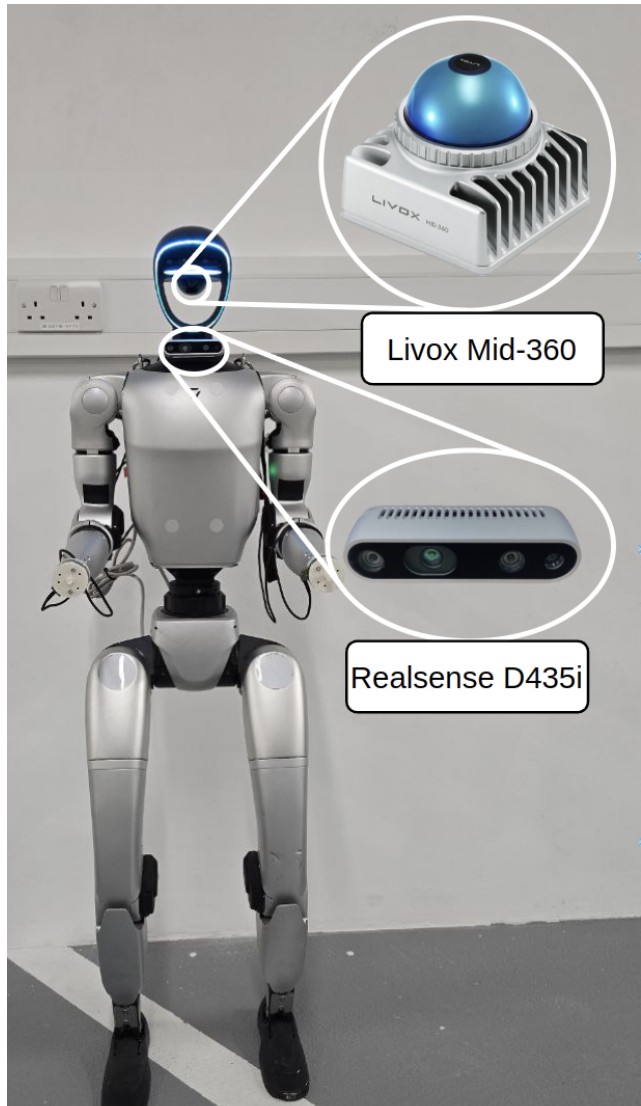

Figure 9: Unitree G1 robotic platform.

### A.4.2 ROBOTIC PLATFORM

For this study, we utilized a Unitree G1 humanoid robot equipped with a Livox Mid-360 LiDAR and a RealSense D435i RGB-D camera. Simultaneous localization and mapping (SLAM), path planning, and low-level control are executed entirely on the robot's onboard computer. In addition to that image-goal navigation process was performed on an external laptop connected to the robot via an Ethernet cable. A photograph of the robotic platform is provided in Figure 9.

### A.4.3 ODOMETRY AND MAPPING

We employed RTAB-Map Labbé & Michaud (2019) as the primary module for pose estimation, mapping, and localization. The robot leverages the onboard Livox Mid-360 LiDAR to compute odometry through RTAB-Map's ICP-based odometry module. Additionally, RGB-D images captured by the RealSense camera are integrated into RTAB-Map's SLAM module, enabling robust localization, mapping, and global loop closure detection.

### A.4.4 Data Collection and GS Reconstruction

To formulate the input for gaussian splatting reconstruction, for each recorded frame, the humanoid robot collect the RGB image, the 7D camera pose (a 3D camera position and a 4D camera quaternion), and the Lidar point cloud. In this real-world setting, we use the Lidar point cloud as the geometric scaffold instead of the depth map recorded by the realsense camera because the recorded depth map contains lots of artifacts and is very inaccurate in a big environment, whereas the Lidar point cloud is more reliable.

After collecting the data, we first merge the Lidar point cloud of all frames into a complete one using the collected camera poses as shown in Figure 8. This point cloud serves as the initialization of the gaussian splatting field, then we optimize the gaussian splatting field as the usual way in Kerbl et al. (2023).

### A.4.5 Safe Path Planning

Safe navigation and obstacle avoidance are achieved using the ROS Navigation Stack Quigley et al. (2009). This framework integrates sensor inputs from LiDAR and RGB-D cameras to generate occupancy grid maps and perform path planning. Specifically, the navigation stack utilizes costmap-based planning algorithms such as Dijkstra's algorithm and the Dynamic Window Approach (DWA) to calculate collision-free paths in real-time, ensuring robust and safe trajectories for the humanoid robot within cluttered indoor environments.

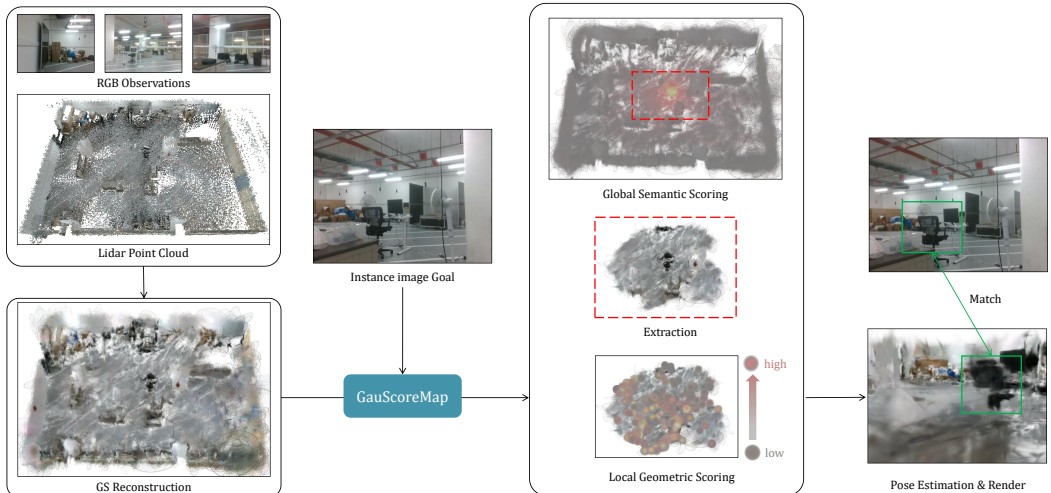

Figure 10: An example of finding the goal position of an instance image goal dipicting a chair with our collected real-world data.

### A.4.6 An example of IIN using our GauScoreMap

We demonstrate an example of locating an instance image goal using our collected real-world data in Figure 10. First, we leverage the merged LiDAR point cloud and calibrated RGB images to reconstruct the Gaussian splatting field of the scene, as shown in the left portion of Figure 10.

An instance image goal depicting a chair is then provided to our GauScoreMap method, which processes it through two sequential scoring stages. The global semantic scoring stage generates a coarse localization map that roughly identifies the chair's position, while the local geometric scoring stage produces a refined location estimate. The top-$k$ scoring rays are subsequently selected to estimate the camera pose and render an image of the target position.

As demonstrated on the right side of Figure 10, the rendered image successfully captures the chair specified in the instance image goal.

| Ablation Choice | SR |
|---|---|
| $\tau = 0.006, K = 10240, k = 100$ (default) | 0.766 |
| $\tau = 0.012$ | 0.700 |
| $\tau = 0.003$ | 0.633 |
| $K = 20480$ | 0.766 |
| $K = 5120$ | 0.700 |
| $k = 200$ | 0.766 |
| $k = 50$ | 0.733 |

Table 5: The influence of some important hyperparameters on the navigation success rate.

Table 6: Locating Success Rate of our method under different portions of gaussian deletion

|  | Delete 0% | Delete 20% | Delete 40% | Delete 60% |
|---|---|---|---|---|
| SR | 100% | 100% | 90% | 75% |

### A.5 ABLATION STUDY OF IMPORTANT HYPERPARAMETERS

We analyze the influence of three key hyperparameters on navigation success rate, testing their impact across 30 randomly selected episodes from the HM3D validation set while holding all other settings unchanged:

- Global Scoring: The threshold $\tau$, used to filter relevant Gaussian candidate regions.

- Local Scoring: $K$, the number of rays sampled per region; and $k$, the number of top-scoring rays selected for final camera pose estimation.

The ablation results is shown in Table 5 below, and the default settings of these 3 hyperparameters are $\tau = 0.006, K = 10240, k = 100$.

Our ablation analysis reveals differential sensitivity to the hyperparameters governing the two-stage scoring mechanism:

Sensitivity to $\tau$ (Global Scoring Threshold): Our method is more sensitive to the global scoring threshold $\tau$. A larger $\tau$ results in less accurate candidate region segmentation, substantially enlarging the search space for the local scoring stage and decreasing efficiency. Conversely, a smaller $\tau$ makes the segmentation less tolerant of partially relevant regions, risking the erroneous filtration of small but correct candidate regions.

Insensitivity to $K$ and $k$ (Local Scoring Parameters): The method exhibits higher sensitivity to $K$ (number of sampled rays) and lower sensitivity to $k$ (number of top rays for pose estimation). As sampled rays are inherently disordered, $K$ must be sufficiently large to reliably cover the relevant rays; using a smaller number (e.g., $K = 5120$) leads to less accurate pose estimation. Furthermore, since a limited number of high-scoring rays is adequate to determine the camera pose, setting $k = 100$ represents an effective trade-off between pose estimation accuracy and computational efficiency.

### A.6 ROBUSTNESS TO INCOMPLETE SCENE EXPLORATION

Since our method relies on a reconstructed GS scene for navigation, we evaluate its robustness to incomplete exploration where target objects may be partially occluded or incompletely reconstructed. We simulate this by manually deleting portions of target object Gaussians to test localization performance under degraded conditions. We tested 20 successful episodes across 5 HM3D Yadav et al. (2023) scenes, progressively deleting around 20%, 40%, and 60% of target region Gaussians. Table 6 and Figure 11 show that our method maintains high localization success even at 40% deletion. This robustness stems from two factors: the semantic scoring accurately identifies candidate objects despite incomplete shapes, while local geometric scoring leverages surrounding texture details to determine the correct camera pose.

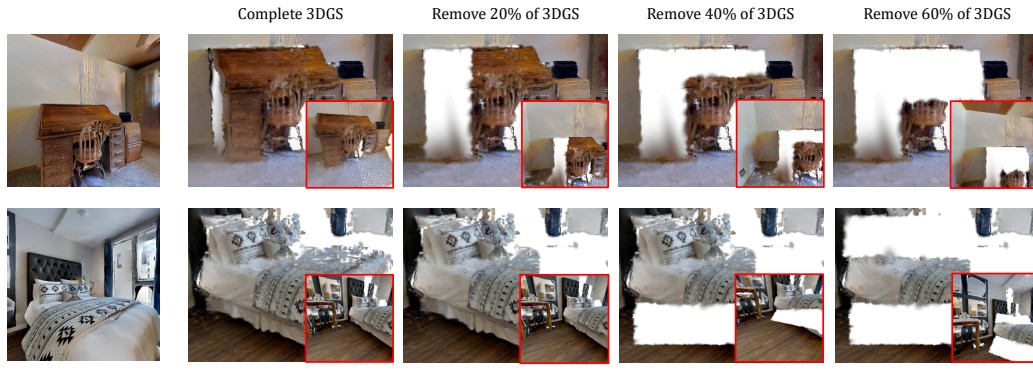

Complete 3DGS    Remove 20% of 3DGS    Remove 40% of 3DGS    Remove 60% of 3DGS

Instance-Image Goal    3DGS Removal and Located Image

Figure 11: The robustness of our method under different portions of gaussian deletion. The red boxed image of the bottom right corner is the located image by our method.

The key reason our method can still localize the target image with such a large portion of gaussian deletion is in the semantic scoring stage: we project the full-image CLIP feature onto the Gaussian scene, assigning each individual Gaussian a complete semantic signal (e.g., "chair," "bed"). As a result, even if more than 90% of an object's Gaussians are removed, the remaining ones still retain enough semantic evidence to be matched by the text feature. Figure 12 demonstrates this extreme test—deleting over 90% of the target Gaussians—showing that the semantic scoring stage continues to highlight the surviving Gaussians with high scores, allowing them to be proposed as valid candidate regions.

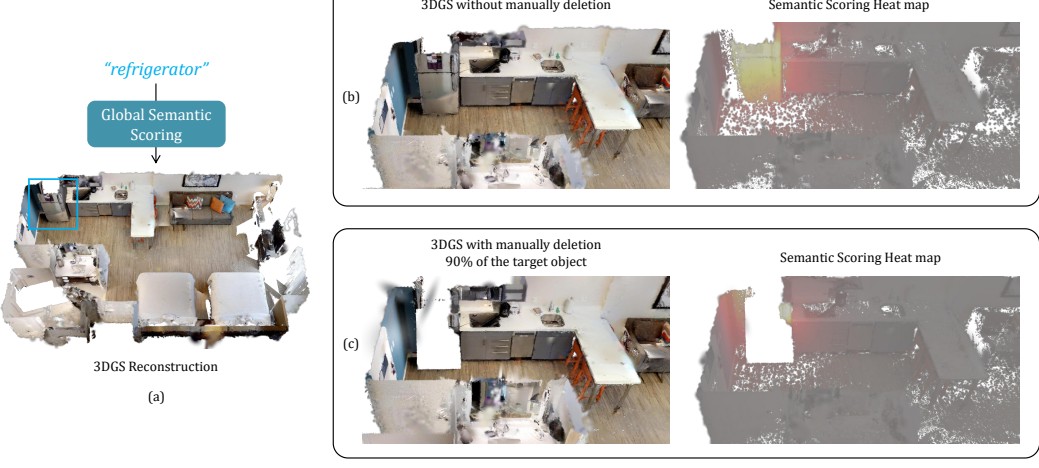

Figure 12: An example of semantically locating a target object after deleting more than 90% of its Gaussians. (a) The semantic scoring stage takes "refrigerator" as the text query and score the full 3DGS. (b) In the complete scene, the scoring map correctly highlights the refrigerator with high scores. (c) Even after removing over 90% of the object's Gaussians, the remaining fragments still receive strong scores, clearly distinguishing them from unrelated surroundings.

## A.7 GAUSSIAN RECONSTRUCTION METRICS FOR HM3D VALIDATION SET

Across the 1,000 evaluation episodes in the HM3D validation set, there are 36 unique scenes. For each scene, we report the key Gaussian-splatting reconstruction statistics, including the number of collected input images, the total reconstruction time, and the average PSNR of the rendered Gaussian outputs relative to the collected images in Table 7 as a reference.

| Scene | Number of Collected Images | Reconstruction Time (min) | PSNR (dB) |
|---|---|---|---|
| 4ok3usBNeis | 564 | 8.7 | 35.07 |
| 5cdEh9F2hJL | 1104 | 17.6 | 36.55 |
| 6s7QHgap2fW | 468 | 6.9 | 34.52 |
| 7MXmsvcQjpJ | 420 | 6.2 | 33.55 |
| a8BtkwhxdRV | 540 | 8.3 | 35.07 |
| BAbdmeyTvMZ | 900 | 13.4 | 34.43 |
| bCPU9suPUw9 | 456 | 6.7 | 36.70 |
| bxsVRursffK | 652 | 9.9 | 33.82 |
| CrMo8WxCyVb | 862 | 12.8 | 34.86 |
| cvZr5TUy5C5 | 864 | 12.9 | 35.50 |
| Dd4bFSTQ8gi | 1188 | 18.6 | 36.45 |
| DYehNKdT76V | 668 | 10.1 | 35.03 |
| eF36g7L6Z9M | 2136 | 31.9 | 33.69 |
| GLAQ4DNUx5U | 864 | 12.9 | 36.53 |
| h1zeeAwLh9Z | 744 | 11.2 | 35.71 |
| HY1NcmCgn3n | 2112 | 31.5 | 33.50 |
| k1cupFYWXJ6 | 600 | 9.2 | 34.46 |
| LT9Jq6dN3Ea | 1164 | 18.3 | 35.90 |
| MHPLjHsuG27 | 228 | 3.5 | 36.86 |
| mL8ThkuaVTM | 276 | 4.2 | 34.41 |
| mv2HUxq2B53 | 1776 | 26.8 | 33.86 |
| Nfvxx8J5NCo | 312 | 4.7 | 36.38 |
| p53SfW6mjZe | 1128 | 17.7 | 33.39 |
| q3zU7Yy5E5s | 708 | 10.7 | 36.10 |
| q5QZSEeHe5g | 444 | 6.6 | 34.83 |
| QaLdnwvtxbs | 696 | 10.5 | 32.60 |
| qyAac8rV8Zk | 420 | 6.2 | 34.61 |
| svBbv1Pavdk | 960 | 14.4 | 35.67 |
| TEEsavR23oF | 336 | 5.1 | 35.48 |
| VBzV5z6i1WS | 708 | 10.7 | 33.80 |
| y9hTuugGdiq | 1380 | 21.3 | 34.86 |
| yr17PDCnDDW | 1380 | 21.3 | 34.19 |
| ziup5kvtCCR | 840 | 12.5 | 33.56 |
| zt1RVoi7PcG | 888 | 13.2 | 36.83 |

Table 7: Gaussian Reconstruction Metrics for HM3D validation set

