# OpenReview forum: "Hierarchical Scoring with 3D Gaussian Splatting for Instance Image-Goal Navigation"
_ICLR.cc/2026/Conference — Submitted to ICLR 2026_

### Official Review · Reviewer_ZawG · 2025-10-29

**Soundness:** 2
**Presentation:** 3
**Contribution:** 2
**Rating:** 4
**Confidence:** 4

**Summary:**

The paper proposes GauScoreMap for Instance Image Navigation (IIN). The method reconstructs a 3D Gaussian Splatting scene, “lifts” CLIP visual features into the 3D domain, then applies a global semantic scoring phase based on class-text embeddings to identify candidate regions, followed by a local geometric scoring phase that uses cross-attention between ray features and target-image DINOv2 features for fine-grained pose estimation. Experiments on the HM3D IIN validation set report the best SR (0.784) and SPL (0.605), supported by ablations, efficiency studies, and limited real-world demonstrations.

**Strengths:**

**Hierarchical scoring reduces redundancy**

* The global–local scoring structure effectively filters candidate regions via CLIP-based text–image similarity before precise geometric refinement, significantly reducing computation and redundancy.

**Empirical advantages over strong baselines**

* On the HM3D IIN benchmark, the method surpasses GaussNav by +5.9% SR and +2.7% SPL, demonstrating consistent gains under the same 3DGS representation.
* The method remains robust when 20–60% of target-region Gaussians are removed, showing resilience to incomplete reconstructions.

**Weaknesses:**

**Mathematical clarity and consistency issues**

* The intermediate formula between Eq. (5) and Eq. (6), ( $\ell = \max((O - v_o) \cdot v_d, , 0)$ ), contains a typographical comma error and unclear notation.

**Strong dependence on object category detection**

* The global stage depends on Mask R-CNN for class labels; failure in detection or class confusion among nearby instances may bias candidate generation, yet no failure or robustness analysis is reported.

**Lack of reproducibility and openness**

* No code or pretrained models are provided, and there is no stated plan or timeline for release.

**Questions:**

* How do these training settings influence convergence speed or stability?
* Have the authors observed any failure patterns when CLIP’s text embeddings misalign with visual attributes?
* Would introducing instance-level image embeddings, rather than purely text-based class embeddings, help reduce bias?
* Are there plans to release the implementation and pretrained weights? If so, when?
* How sensitive is the performance to Gaussian count, rendering resolution, or backbone selection (e.g., DINOv2 vs. CLIP)?
* Beyond empirical improvement, what conceptual or theoretical insight does GauScoreMap contribute regarding representation alignment between 2D semantics and 3D geometry?
* How does this approach advance understanding of the trade-off between global semantic priors and local geometric consistency in 3D neural fields?
* Could the authors share any empirical or theoretical insights explaining why this two-stage design improves over a unified cross-modal model?
* Finally, could you explicitly articulate the motivation behind the hierarchical scoring and the insights gained?

---

> ### Author Response · Authors · 2025-11-19
> **Response (Part 1)**
>
> We thank the Reviewer for the time devoted to our paper, providing insightful feedback and for appreciating our contributions. In the following, we address the concerns and questions pointed out by the Reviewer.
>
> 1. **Mathematical clarity and consistency issues: The intermediate formula between Eq. (5) and Eq. (6), contains a typographical comma error and unclear notation.**
>
>    We are sorry for the typo. We have already fixed the typo in the revised paper.
>
>
>
> 2. **Strong dependence on object category detection: The global stage depends on Mask R-CNN for class labels; failure in detection or class confusion among nearby instances may bias candidate generation, yet no failure or robustness analysis is reported.**
>
>    We thank the reviewer for pointing this out. We agree that the failure in detection or class confusion is one cause of the navigation failure. And we have conducted a failure case analysis below. As it's time-consuming for us to analyze failure cases for all evaluation episodes, we just randomly pick 100 episodes for such failure analysis. The evaluated success rate is 80%, so for the failed 20 cases, we analyzed the reasons for their failures:
>
>    | Failed Reasons                       | Times/Percentage |
>    | ------------------------------------ | ---------------- |
>    | Wrong category detection             | 8/40%            |
>    | Failed at picking the correct region | 4/20%            |
>    | Failed at pose estimation            | 8/40%            |
>
>    **Failure Mode Analysis:**
>
>    1. **Wrong category detection** (frequent for "plant" and "television"): Small plants appear fragmented and blurry in HM3D scans, leading to the CLIP encoder fail to encode the right semantic embedding to it, and the Mask-RCNN also fails to output "plant" or "flower" for the target images due to the broken scans. Televisions appear as dark flat surfaces without a complete shape in the HM3D scans, hindering accurate semantic encoding.
>    2. **Failed at picking the correct region** (common for "couch"): Multiple candidate regions with similar texture patterns yield minimal score differences between correct/incorrect regions, causing the method fail to pick the right candidate gaussian region.
>    3. **Failed at pose estimation**: Final localized image doesn't face the target or remains too distant. Requires more sophisticated local alignment methods for better position refinement.
>
>
>
> 3. **Lack of reproducibility and openness: No code or pretrained models are provided, and there is no stated plan or timeline for release.**
>
>    We thank the reviewer for the helpful comment. We have provided the full technical details in the main paper to ensure reproducibility and will add extra ablation experiments to show the influence of each hyperparameters in the revised version, and we will release the complete implementation upon acceptance.

---

> > ### Author Response · Authors · 2025-11-19
> > **Response (Part 2)**
> >
> > 4. **How do these training settings influence convergence speed or stability?**
> >
> >    We thank the reviewer for raising this question. There are 2 stages of training in our pipeline, one is the the gaussian splatting scene training and the other is the local score function training. For the gaussian splatting scene training stage, since it is fully-established in the previous work MAGIC-SLAM, we just adopt to its default setting, which maintains a fast convergence speed and stability as shown in Table 3. For the local score function training, we have conducted extra experiments on how the training settings will influence the convergence speed and final performance, all settings will converge within 15 minutes in a RTX 4090 GPU.
> >
> >    We analyze the influence of three key hyperparameters on navigation success rate, testing their impact across 30 randomly selected episodes from the HM3D validation set while holding all other settings constant:
> >
> >    - **Global Scoring:** The threshold $\tau$, used to filter relevant Gaussian candidate regions.
> >    - **Local Scoring:** $K$, the number of rays sampled per region; and $k$, the number of top-scoring rays selected for final camera pose estimation.
> >
> >    | Ablation Choice                        | SR    |
> >    | -------------------------------------- | ----- |
> >    | $\tau = 0.006,K=10240,k=100$ (default) | 76.6% |
> >    | $\tau = 0.012$                         | 70.0% |
> >    | $\tau = 0.003$                         | 63.3% |
> >    | $K=20480$                              | 76.6% |
> >    | $K=5120$                               | 70.0% |
> >    | $k=200$                                | 76.6% |
> >    | $k=50$                                 | 73.3% |
> >
> >    Our ablation analysis reveals differential sensitivity to the hyperparameters governing the two-stage scoring mechanism:
> >
> >    - **Sensitivity to $\tau$ (Global Scoring Threshold):** Our method is more sensitive to the global scoring threshold $\tau$. A **larger $\tau$** results in less accurate candidate region segmentation, substantially enlarging the search space for the local scoring stage and decreasing efficiency. Conversely, a **smaller $\tau$** makes the segmentation less tolerant of partially relevant regions, risking the erroneous filtration of small but correct candidate regions.
> >    - **Insensitivity to $K$ and $k$ (Local Scoring Parameters):** The method exhibits higher sensitivity to $K$ (number of sampled rays) and lower sensitivity to $k$ (number of top rays for pose estimation). As sampled rays are inherently disordered, $K$ must be sufficiently large to reliably cover the relevant rays; using a smaller number (e.g., $K=5120 $) leads to less accurate pose estimation. Furthermore, since a limited number of high-scoring rays is adequate to determine the camera pose, setting $k=100$ represents an effective **trade-off between pose estimation accuracy and computational efficiency**.
> >
> >
> >
> > 5. **Have the authors observed any failure patterns when CLIP’s text embeddings misalign with visual attributes?**
> >
> >    We thank the reviewer for raising this question. One major misalignment in the tested scenes is that with the input of the "chair" text embedding, CLIP feature field scoring will also filter the "sofa" out. This might be because that during the training of CLIP, the "sofa" is also treated as a subcategory of "chair", however, this will not cause a major failure in our final navigation success rate, since the local scoring stage is able to differentiate "chair" and "sofa" in the texture/geometric space.
> >
> >
> >
> > 6. **Would introducing instance-level image embeddings, rather than purely text-based class embeddings, help reduce bias?**
> >
> >    We thank the reviewer for raising this question. We initially explored directly computing the similarity score between the **target instance-level image** and the **3D Gaussian Splatting (GS) feature field** to identify candidate regions. However, this approach yielded poor results because image features generated by the CLIP encoder often contain **inconsistent semantic information** due to **background clutter** (e.g., a chair image also containing a table or sofa). This semantic confusion leads to unreliable scoring and filtering. Consequently, we utilize an object detector (Mask-RCNN) to robustly extract the precise **semantic class label** of the target object, which is then used for global semantic scoring.
> >
> >
> >
> > 7. **Are there plans to release the implementation and pretrained weights? If so, when?**
> >
> >    Sure, we will release the implementation upon accepted.

---

> > > ### Author Response · Authors · 2025-11-19
> > > **Response (Part 3)**
> > >
> > > **8. How sensitive is the performance to Gaussian count, rendering resolution, or backbone selection (e.g., DINOv2 vs. CLIP)?**
> > >
> > > We thank the reviewer for raising this question. Our method is not very sensitive to gaussian count, but sensitive to rendering resolution within a reasonable range, we have conducted an ablation experiment of gaussian count and rendering resolution on the navigation success rate below:
> > >
> > > We manually adjust sparsify the gaussians of the reconstructed scenes to ablate the performance change led by gaussian count $N_G$; and since we do not render full-resolution images for local scoring stage, instead, we only sample a batch of rays for matching, we treat "rendering resolution" here as the number of sampled rays $K$ for local scoring stage. We keep all the other settings unchanged, and evaluate the navigation success rates for 30 randomly chosen episodes from the HM3D evaluation set.
> > >
> > > | Ablation Choice                                        | SR    |
> > > | ------------------------------------------------------ | ----- |
> > > | $N_G\approx$18,000,000 on average; $K=10240$ (default) | 76.6% |
> > > | $K=20480$                                              | 76.6% |
> > > | $K=5120 $                                              | 70.0% |
> > > | $K=2560 $                                              | 66.6% |
> > > | $N_G \approx 18,000,000 * 0.75=13,500,000$             | 76.6% |
> > > | $N_G \approx 18,000,000 * 0.5=9,000,000$               | 76.6% |
> > > | $N_G \approx 18,000,000 * 0.25=4,500,000$              | 76.6% |
> > >
> > > The table demonstrates that **sparsifying the Gaussian Splatting representation down to 25% of the original count does not significantly affect the navigation success rate.** This resilience suggests a high degree of **redundancy** in the dense Gaussian representation.
> > >
> > > Our method is inherently **insensitive to the total Gaussian count** because both key stages operate on a small subset:
> > >
> > > - **Semantic Scoring:** Only requires $\approx 600,000$ Gaussians (as discussed in Section 4.4) to accurately identify target semantic object regions.
> > > - **Local Scoring:** Relies on a small, fixed number of **Gaussian-ray pair samples** (defaulting to $10,240$).
> > >
> > > Conversely, our method is **sensitive to the number of sampling rays (rendering resolution).** To ensure high accuracy, we must sample enough rays to guarantee the selection of at least 100 high-scoring rays necessary for precise target localization. The default number of $10,240$ rays is chosen as an effective **trade-off between final localization quality and computational efficiency.**

---

> ### Author Response · Authors · 2025-11-19
> **Response (Part 4)**
>
> (following response of question 8)
>
> We have also conducted the ablation experiment of using different backbone both stages.
>
> In the first semantic scoring stage, the candidate gaussian regions need to be extracted by comparing text feature and visual feature, therefore, we can only use CLIP (or other CLIP-like Contrastive Language-Image Pre-training networks) instead of DINOv2 as the plain DINOv2 does not support comparison between text feature and visual feature. Other than CLIP, we have found that SigLIP can also serve as a better alternative, so we replace CLIP with SigLIP to compare the performance.
>
> And in the second geometric scoring stage, we need a pre-trained visual feature extraction backbone to enhance the extracted visual features, therefore, we use DINOv2 which is commonly used as the strong image feature extraction backbone in image classification, segmentation, vision-language-action models and more. And it is true that the image encoder of CLIP can also be used as an alternative to DINOv2 to extract features in the geometric scoring stage, so we conduct the ablation of using the DINOv2 or CLIP image encoder in the geometric scoring stage to see the influence of the backbone. Actually, a CNN-based feature extraction network can also be used as the backbone, therefore, we choose the image encoder of SuperPoint to also replace the original DINOv2 to compare the performance.
>
> We randomly choose 30 episodes from the HM3D validation episodes, keeping the left modules unchanged but only change the backbone of the semantic scoring stage or the backbone of the local scoring stage, to see how does this change influence the navigation success rate and the related computation time:
>
> | **Backbone of Semantic Scoring** | **SR** | Feature Uplifting time of Semantic Scoring Stage |
> | -------------------------------- | ------ | ------------------------------------------------ |
> | CLIP (default)                   | 76.6%  | 1.12s                                            |
> | SigLIP                           | 76.6%  | 1.12s                                            |
> | **Backbone of Local Scoring**    | **SR** | **Training time of Local Scoring Stage**         |
> | DINOv2 (default)                 | 76.6%  | 15 min                                           |
> | CLIP image encoder               | 76.6%  | 33 min                                           |
> | SuperPoint image encoder         | 70.0%  | 10 min                                           |
>
> We observed that **changing the backbone** in the semantic scoring stage (from CLIP to SigLIP) resulted in **no noticeable change** in either the **navigation success rate** or the **feature extraction time**. This is because both CLIP and SigLIP variants with the same architecture (e.g., both using a similar ViT image encoder) offer **comparable feature extraction performance and speed** during inference and identifying the six target object categories within the HM3D validation set does not present a **critical challenge** that only SigLIP can resolve.
>
> The **local scoring stage** showed **identical success episodes** when the image encoder was changed from **DINOv2** to **CLIP**, confirming that the **CLIP image encoder** also serves as an effective backbone for extracting image features.
>
> However, using CLIP **doubled the training time** compared to DINOv2. This performance slowdown is primarily due to the difference in how image-level features are generated:
>
> - **CLIP:** By default, the standard CLIP encoder generates a **single, global embedding** for the entire image. To obtain a dense, **image-level feature map** required for local scoring (like the approach in LERF), we had to employ a **sliding window method**. This process repeatedly extracts a feature for a small patch, which is computationally expensive and time-consuming.
> - **DINOv2:** This model inherently supports the extraction of dense, high-resolution feature maps across the entire image in a single, efficient forward pass, resulting in significantly **faster training**.
>
> Consequently, we selected **DINOv2** as the default encoder for its strong performance and superior training efficiency. Conversely, the **SuperPoint** features resulted in a **decrease in performance**, indicating that its extracted features are less robust than those from DINOv2 and CLIP for this task.

---

> > ### Author Response · Authors · 2025-11-19
> > **Response (Part 5)**
> >
> > 10. **How does this approach advance understanding of the trade-off between global semantic priors and local geometric consistency in 3D neural fields?**
> >
> >     We thank the reviewer for their question. We advance the understanding of the trade-off between **global semantic priors** and **local geometric consistency** by showing a successful strategy for their **hierarchical utilization** within our system.
> >
> >     Our **instance-image goal navigation (IIN)** task fundamentally requires both: the **Global Semantic Prior** is needed for **initial, efficient filtering** of the scene based on the object's general category (scope), while **Local Geometric Consistency** is mandatory for the **final, precise verification** of the specific instance's unique visual features (certainty).
> >
> >     Our **2-stage hierarchical scoring method** addresses the typical conflict between these two information types by **decoupling** them into a strategic sequence. Stage 1 leverages the efficiency of the semantic prior for coarse, large-scale search space reduction. Stage 2 then relies on the precision of local geometric consistency, captured by the reconstructed scene, for fine-grained instance matching and definitive identification. This modular approach provides an empirical model showing that the optimal solution is not a compromise, but a **strategic sequence** utilizing each information type's unique strength.
> >
> >
> >
> > 11. **Could the authors share any empirical or theoretical insights explaining why this two-stage design improves over a unified cross-modal model?**
> >
> >     We thank the reviewer for their question. We have answered together in detail in Q12 below.
> >
> >
> >
> > 12. **Finally, could you explicitly articulate the motivation behind the hierarchical scoring and the insights gained?**
> >
> >     We thank the reviewer for their question. Our method is strongly driven by the requirements of the **instance-image goal navigation** task. Unlike **object-goal** navigation (finding a semantic category) or **image-goal** navigation (finding a texture-level match), instance-image goal requires finding a **specific instance** of an object shown in the goal image.
> >
> >     This dual requirement—identifying the **semantic meaning** of the object and then matching its **exact instance/texture**—motivates our **2-stage hierarchical scoring method**:
> >
> >     1. **Stage 1:** Filter the scene using the **semantic meaning** extracted from the goal image.
> >     2. **Stage 2:** Perform finer matching and comparison using the reconstructed scene, which contains rich **texture and geometric information**.
> >
> >     Compared to a unified model, the first stage of semantic extraction can narrow down the search space by a large portion, making it easier for the geometric scoring stage to find the right pose. Without the 2-stage design, a unified model will search in a very large scene to compare all areas with the image, which will potentially cause longer search time and redundant 2D-3D comparisons.

---

> > > ### Comment · Reviewer_ZawG · 2025-11-27
> > >
> > > I will keep my current recommendation for now. However, I am open to raising my score if the authors can clarify the following points.
> > >
> > > 1. After reading Reviewer YJ7j’s review, I now realize that your method requires a first exploratory rollout to build the scene representation. Given that you already have a fairly complete exploration of the environment, why is it not feasible to use existing 3D instance or semantic segmentation pipelines to pre compute a 3D or 2D semantic map, and then directly run a classical planner such as A* to find the optimal path? My impression is that there are many stronger and faster 3D instance or semantic segmentation solutions in the literature than a CLIP plus Mask R CNN pipeline. It would help if you could explain why such alternatives are not competitive or not applicable in your setting.
> > >
> > > 2. In the first stage, when you use CLIP based semantic scoring to filter candidate regions, how often does this stage incorrectly drop the true target region? Can you quantify how many final failures are directly caused by such false negatives in the semantic filtering stage?
> > >
> > > 3. I still do not fully understand the motivation behind the hierarchical scoring design. Conceptually, it still feels quite similar to the original brute force search, but with a more efficient scoring procedure over viewpoints. I do not think that a “small” improvement cannot be an ICLR contribution. However, I would like a clearer explanation of what is the key conceptual gain beyond engineering efficiency, and why this view of the method is the most appropriate way to understand your contribution.

---

> ### Author Response · Authors · 2025-11-27
> **Response (Part 1)**
>
> We thank the reviewer for your carefully reading our response and also our response for other reviewers. And we are very willing to answer your questions below:
>
> 1. **After reading Reviewer YJ7j’s review, I now realize that your method requires a first exploratory rollout to build the scene representation. Given that you already have a fairly complete exploration of the environment, why is it not feasible to use existing 3D instance or semantic segmentation pipelines to pre compute a 3D or 2D semantic map, and then directly run a classical planner such as A-star to find the optimal path? My impression is that there are many stronger and faster 3D instance or semantic segmentation solutions in the literature than a CLIP plus Mask R CNN pipeline. It would help if you could explain why such alternatives are not competitive or not applicable in your setting.**
>
>    We thank the reviewer for this insightful question.
>
>    For the question of "Why not use existing 3D instance or semantic segmentation pipelines to pre-compute a 3D/2D semantic map and then run A*":
>
>    - Our task is **instance image–goal navigation**, not object-goal navigation. Thus, the agent must not only reach an object of the correct semantic category but must locate **the exact object instance** shown in the query image (e.g., *that specific chair*, not any chair).
>
>    - And a purely semantic 3D map only provides **category-level** information. In contrast, identifying the correct instance requires **fine-grained geometric and appearance matching**, which motivates our **local geometric scoring stage**. Once the instance location is identified, any classical planners such as A* can indeed be used.
>
>    - Although a 3D instance segmentation method can differentiate multiple instances of the same category (e.g., “chair_1,” “chair_2”), **these labels cannot be matched to the instance detected in the 2D goal image**. The reason is that a 2D instance detector only outputs a category label (“chair”) and assign it a random instance index to make it "chair_x" (x might be 0,1,2,...), this index is not consistent with the one detected using the 3D instance segmentation method. Thus, even if we assign unique instance IDs in 3D, the 2D detector provides no ensurance to determine whether the goal image corresponds to “chair_1” or “chair_2” in the 3D scene. Because the 2D and 3D pipelines do not share a unified instance identity space, their outputs cannot be aligned, making 3D instance labels unusable for our instance-image navigation setting.
>
>
>    For the question of "Why use CLIP features + Mask R-CNN instead of a full 3D instance/semantic segmentation pipeline?":
>
>    - **CLIP embeddings are more flexible than hard semantic labels**. Existing 3D semantic pipelines such as GaussNav assign **one fixed label** to each 3D element. This is rigid in practice. For example, a potted plant with flowers may be labeled “plant” in the 3D map, while Mask R-CNN may detect the target image as “flower.” Under hard labels, this region would never be selected as a candidate. Using **CLIP features** soft-encodes semantics: both “plant” and “flower” queries activate the same region because CLIP’s embedding space captures cross-category similarity. This greatly improves robustness and avoids brittle failure cases caused by inconsistent or ambiguous class labels.
>
>    - **Soft CLIP features preserve useful context for geometric localization.** In GaussNav (which uses the reviewer-suggested pipeline), semantic labels create **very sharp boundaries** around each object. When filtering for the target class (e.g., “chair”), only the chair’s Gaussians remain. However, **localizing a camera pose from a single image requires contextual background information**—walls, floor, neighboring furniture, etc. As shown in Figures 3–7 of our paper, our CLIP-filtered candidate regions naturally include both the target object and the surrounding Gaussians that appear in the target image. These contextual Gaussians are crucial: many of the top-k matching rays originate on background geometry. Without them, the camera pose becomes under-determined, and the system must resort to expensive multi-view rendering and sampling **(30–50 s in GaussNav**, as reported), compared to our method (**within 1.5s**). Thus, CLIP-based soft filtering **enables faster and more accurate instance localization**, because it preserves the contextual cues required for fine-grained geometric matching.

---

> > ### Author Response · Authors · 2025-11-27
> > **Response (Part 2)**
> >
> > 2.**In the first stage, when you use CLIP based semantic scoring to filter candidate regions, how often does this stage incorrectly drop the true target region? Can you quantify how many final failures are directly caused by such false negatives in the semantic filtering stage?**
> >
> > We thank the reviewer for raising this question. In practice, this “incorrectly dropping the true target region” issue rarely occurs for the four major categories {chair, couch, bed, toilet}, but it does occasionally appear for {television, plant}. This observation is consistent with the failure analysis provided in our response to Q2. Within the limited rebuttal period, we randomly sampled 10 episodes per category and inspected the semantic scoring stage to check whether the true target region was incorrectly filtered out. The results are summarized in the table below.
> >
> > | Category   | Success in finding the target region | Incorrectly drop the target region | Total Episodes |
> > | ---------- | ------------------------------------ | ---------------------------------- | -------------- |
> > | chair      | 10                                   | 0                                  | 10             |
> > | couch      | 10                                   | 0                                  | 10             |
> > | bed        | 10                                   | 0                                  | 10             |
> > | toilet     | 10                                   | 0                                  | 10             |
> > | television | 7                                    | 3                                  | 10             |
> > | plant      | 8                                    | 2                                  | 10             |
> >
> > From these success and failure cases, we draw the following conclusion:
> >
> > - **The quality of the HM3D scan is a dominant factor affecting semantic scoring reliability.** The categories {chair, couch, bed, toilet} correspond to large, well-scanned objects with complete geometry and distinctive textures. As a result, both Mask R-CNN and CLIP can reliably extract the correct semantic signals. In contrast, for {television, plant}, the HM3D reconstructions are often highly incomplete, with objects appearing as fragmented or broken pieces. This severely degrades the semantic cues available in the 3DGS, making it harder for either model to recover the correct semantic evidence, which leads to occasional drops of the true target region.

---

> > > ### Author Response · Authors · 2025-11-27
> > > **Response (Part 3)**
> > >
> > > 3.**I still do not fully understand the motivation behind the hierarchical scoring design. Conceptually, it still feels quite similar to the original brute force search, but with a more efficient scoring procedure over viewpoints. I do not think that a “small” improvement cannot be an ICLR contribution. However, I would like a clearer explanation of what is the key conceptual gain beyond engineering efficiency, and why this view of the method is the most appropriate way to understand your contribution.**
> > >
> > > We thank the reviewer for this important question. The motivation behind the hierarchical scoring design is not merely efficiency; it arises directly from the structure of the **Instance Image Goal Navigation** task and the unique properties of the 3DGS representation. We clarify the conceptual motivation below.
> > >
> > > - The task itself naturally decomposes into two levels of information. In this task, the input is an image containing a specific object instance. Solving the problem requires leveraging **two distinct types of information**:
> > >
> > >   - **Coarse, semantic information**: identifying objects in the scene that share the same category as the query image (e.g., “chair”).
> > >   - **Fine, appearance-level information**: matching the **exact instance** using detailed texture and background cues present in the image.
> > >
> > >   This decomposition is intrinsic to the problem: no single mechanism can simultaneously perform global semantic filtering and fine-level geometric/appearance matching efficiently. Thus, a **coarse-to-fine, divide-and-conquer formulation** is a natural conceptual structure rather than an engineering trick.
> > >
> > > - 3D Gaussian Splatting provides **a unified representation** that supports both the coarse semantic-level stage and the fine texture-level stage.
> > >
> > >   - For the coarse stage: Each Gaussian explicitly stores position and can be augmented with semantic embeddings. Encoding CLIP features in the Gaussians transforms the entire scene into a soft, expressive semantic field, enabling robust category-level filtering even under imperfect detection (as detailed in Response 1).
> > >   - For the fine stage: Unlike point clouds or voxel maps, 3DGS is inherently designed for **efficient view- and ray-based rendering**. Because the representation can produce view-dependent color information, it allows **direct comparison between 3D appearance and the query 2D image**. And we improve the comparison efficiency further with our local geometric scoring by replacing the redundant many-view sampling in GaussNav to only a small batch of ray sampling, improving the matching speed from 30-50s in GaussNav to only less than 1.5s.
> > >
> > >   Thus, the hierarchical design emerges directly from leveraging two complementary strengths of 3DGS within one framework.
> > >
> > > - Why this is **conceptually different from brute-force search**:
> > >
> > >   GaussNav performs a brute-force search at the fine stage: it renders dozens of full views around candidate regions and selects the best match.
> > >
> > >   In contrast, our hierarchical formulation changes the nature of the problem:
> > >
> > >   - After the semantic stage, we restrict attention to a **soft, semantically coherent region**, not a set of disconnected object labels.
> > >   - Our local scoring stage does **not** search over views; instead, it searches over **rays**, leveraging the geometric constraints within highly-relevant rays to locate the 6D camera pose. This localization method leverages both the geometric and texture information of the representation to efficiently locate the camera pose, which has a clear geometric guidence and is sample-efficient.
> > >   - This design reduces the fine-level matching from **30–50 seconds** (GaussNav) to **under 1.5 seconds** while also increasing robustness, which is at least 20 times faster than the brute-force search view-matching method.
> > >
> > >   The key conceptual gain is that the problem is reframed from “render many views and compare them” to “match the 2D image at the ray level using the intrinsic properties of the 3D representation.” This is not simply an engineering acceleration but a **different structural decomposition** of the navigation problem.

---

> > > > ### Comment · Reviewer_ZawG · 2025-11-27
> > > >
> > > > I appreciate the detailed responses so far and I have one remaining conceptual confusion that I would like to clarify.
> > > >
> > > > 1. In your new analysis, you show that the CLIP based global semantic scoring stage rarely drops the true target region for the main four categories, and only occasionally fails for {television, plant}. This suggests that the global semantic stage has quite high recall at the level of candidate regions. However, in the ablation study, the variant “GauScoreMap w/o Local Geometric Scoring” only achieves SR ≈ 0.42, which is much lower than the full model. Could you explain more concretely why there is such a large performance gap if the CLIP based global semantic stage already covers the true target region in most episodes? In other words, how much of this gap comes from failures of the global semantic stage to include the true region at all, and how much comes from the inability of the global stage alone to select the correct instance and viewpoint among the remaining candidates?
> > > >
> > > > 2. Related to this, I would be very interested in the following thought experiments or ablations that ignore efficiency.
> > > >
> > > >    * If you take the CLIP based global semantic map as a scoring function and perform a more exhaustive or brute force search over viewpoints (for example, evaluating many more candidate viewpoints directly using the semantic map), what SR would you obtain if you do not constrain computation time?
> > > >    * Conversely, if you replace the global semantic stage with the local geometric scoring network and then perform a brute force search over viewpoints using only the local scores, again without counting efficiency, how would the SR change?
> > > >      These variants would help clarify whether the main gains come from the hierarchical structure itself, from the scoring functions, or from the way you restrict the search space.
> > > >
> > > > 3. Regarding the motivation behind the hierarchical scoring, I appreciate your explanation that it follows the coarse to fine structure of the task and leverages 3DGS for both semantic and geometric reasoning. However, at the moment this still reads mostly as a justification of the current architecture in terms of efficiency and task decomposition, rather than as a clearly articulated new problem formulation or conceptual insight. If there is a deeper conceptual perspective here for instance a more general way to view ray level scoring or 3DGS as a joint semantic geometric field I would encourage you to make that more explicit in the final version.
> > > >
> > > > Given that you have already provided quite detailed responses and additional experiments, I am willing to temporarily increase my score at this stage. However, if the final answers or experiments leave me unconvinced about the interpretation of the results, I may revert to my original score in the final decision.

---

> > > > > ### Author Response · Authors · 2025-11-29
> > > > > **Response (Part 1)**
> > > > >
> > > > > We sincerely thank the reviewer for the thoughtful and in-depth discussion, which has greatly helped us clarify both the strengths and limitations of our work. We also deeply appreciate the time and care the reviewer has devoted to reading and engaging with our responses. We address the remaining question below:
> > > > >
> > > > > 1.**In your new analysis, you show that the CLIP based global semantic scoring stage rarely drops the true target region for the main four categories, and only occasionally fails for {television, plant}. This suggests that the global semantic stage has quite high recall at the level of candidate regions. However, in the ablation study, the variant “GauScoreMap w/o Local Geometric Scoring” only achieves SR ≈ 0.42, which is much lower than the full model. Could you explain more concretely why there is such a large performance gap if the CLIP based global semantic stage already covers the true target region in most episodes? In other words, how much of this gap comes from failures of the global semantic stage to include the true region at all, and how much comes from the inability of the global stage alone to select the correct instance and viewpoint among the remaining candidates?**
> > > > >
> > > > > We thank the reviewer for this insightful question.
> > > > >
> > > > > The low SR of the **“GauScoreMap w/o Local Geometric Scoring”** variant arises primarily from the semantic stage’s inability to identify the **correct object instance**, as shown in our ablation study. Our navigation pipeline operates in three steps:
> > > > >  (1) the semantic scoring stage filters the scene into several candidate regions based on the semantic meaning extracted from the target image;
> > > > >  (2) the local geometric scoring stage evaluates these regions and selects the one with the highest geometric score;
> > > > >  (3) ray triangulation within the selected region produces the final 6D target pose.
> > > > >
> > > > > Without the local geometric scoring stage, the agent has **no mechanism** to determine which candidate region corresponds to the exact instance shown in the target image. In typical HM3D indoor scenes, there are often multiple chairs, couches, or beds. Since all belong to the same semantic category, the semantic stage alone cannot differentiate among them. To still report an SR value in this ablation, we let the agent **randomly choose** one of the candidate regions when multiple exist.
> > > > >
> > > > > Moreover, even if the agent **by chance** chooses the correct semantic region, the semantic stage by itself cannot estimate an accurate 6D pose. This often results in the agent facing the wrong direction or stopping too far from the object, leading to failure cases.
> > > > >
> > > > > As we noted in Q2 of our previous response, the semantic scoring stage is generally robust and rarely misses the true target region, except in cases involving severely incomplete or broken 3D scans. Our experiment table confirms that incorrectly dropping the true target region is uncommon.
> > > > >
> > > > > **Regarding the reviewer’s question**—*“how much of the performance gap comes from failures of the global semantic stage to include the true region, and how much comes from the inability of the global stage alone to select the correct instance and viewpoint among the remaining candidates”*—we first clarify that the **global semantic scoring stage is not designed to select the correct instance or viewpoint**, as explained above. Its role is only to retrieve all regions consistent with the target category, not to determine which instance or which camera pose corresponds to the goal image.
> > > > >
> > > > > To provide the deeper error analysis requested by the reviewer, we expand upon our Q2 analysis from our first response. In the table below, the category **“Wrong category detection”** corresponds to the reviewer’s first error type—**“failures of the global semantic stage to include the true region.”** As discussed earlier, this error cannot be attributed solely to the semantic scoring stage, since in cases where the 3D scan is incomplete or broken, Mask R-CNN itself may fail to assign the correct category label.
> > > > >
> > > > > To address the reviewer’s second error type—**“the inability of the global stage alone to select the correct instance and viewpoint among the candidates”**—we run an experiment where we **remove the Local Geometric Scoring Stage** and evaluate whether the semantic stage alone can successfully navigate to the correct instance. Because analyzing all evaluation episodes is time-consuming, we randomly sampled 100 episodes. Given that the full model achieves an 80% SR, we manually analyzed the **20 failure cases** from this subset and categorized their causes as shown in the table below.

---

> ### Author Response · Authors · 2025-11-29
> **Response (Part 2)**
>
> (following the Response of Q1 above)
>
>  | Failed Reasons                       | Failed Times/Percentage |
>    | ------------------------------------ | ----------------------- |
>    | **Wrong category detection**         | **8/40%**   (8 of 20)            |
>    | Failed at picking the correct region | 4/20%  (4 of 20)                 |
>    | Failed at pose estimation            | 8/40%  (8 of 20)                 |
>    | **w/o Local Scoring Stage**          | **67/67%**  (67 of 100)            |
>
> ​We bold the key entries in the table above to improve clarity. From the results, we observe that among the 20 analyzed failure cases, **40%** fall under 	the category of *“failures of the global semantic stage to include the true region.”* As discussed earlier, these errors are primarily due to incomplete or 	severely degraded 3D scans of categories such as **television** and **plant**, which cause Mask R-CNN and the semantic scoring stage to misidentify their 	semantic labels.
>
> ​For the second category—*“the inability of the global stage alone to select the correct instance and viewpoint among the remaining candidates”*—we disable 	the local scoring stage for all 100 sampled episodes. Under this setting, the failure rate rises to **67%**, indicating that the semantic scoring stage by 	itself cannot identify the correct instance or the correct viewpoint, even though it is generally capable of including the true region (as shown in our 	earlier Q2 analysis). Thus, the majority of failures in this setting arise from the absence of the local geometric scoring stage, rather than from the 	semantic stage failing to retrieve the true target region.
>
> 2.**Related to this, I would be very interested in the following thought experiments or ablations that ignore efficiency.**
>
> - **If you take the CLIP based global semantic map as a scoring function and perform a more exhaustive or brute force search over viewpoints (for example, evaluating many more candidate viewpoints directly using the semantic map), what SR would you obtain if you do not constrain computation time?**
> - **Conversely, if you replace the global semantic stage with the local geometric scoring network and then perform a brute force search over viewpoints using only the local scores, again without counting efficiency, how would the SR change? These variants would help clarify whether the main gains come from the hierarchical structure itself, from the scoring functions, or from the way you restrict the search space.**
>
> We thank the reviewer for the helpful suggestions. We have conducted the requested experiments and report the details below.
>
> **For the first setting**, we uniformly sample **[100,300,500,1000] viewpoints** by randomly placing 6-DoF camera poses within the 3D bounding box of each candidate region. For each sampled viewpoint, we render an image and measure its similarity to the target image using **LightGlue**, with the default matching threshold used in the baseline IEVE. We compute the number of matched keypoints between each sampled image and the target image, and the viewpoint with the highest number of matches is treated as the predicted target pose for this brute-force method.
>
> **For the second setting**, the reviewer asks how performance would change if we retained only the local geometric scoring stage and performed a brute-force search over viewpoints. We clarify that the **local geometric scoring stage is not a viewpoint-sampling method**; it operates by sampling **Gaussian–ray pairs**, not by enumerating camera poses. Therefore, it cannot directly perform a “search over viewpoints.” To approximate an equivalent brute-force mechanism, we run **[100,300,500,1000] independent Gaussian–ray sampling rounds** for each episode. In each round, we sample the same number of rays ($K = 10240$) as in our original implementation, since increasing $K$ does not improve single-round accuracy and decreasing $K$ reduces accuracy (as shown in Q8 of our first-round response). Across these [100,300,500,1000] rounds, we select the result whose triangulated camera pose yields the **highest number of LightGlue matches** with the target image, and treat this as the predicted target pose for this brute-force variant.
>
> Due to the time limit of this discussion period, we randomly choose 50 episodes from the HM3D evaluation set, and report the navigation SR of our original implementation, and the brute-force search methods of the above 2 settings in the following table:

---

> > ### Author Response · Authors · 2025-11-29
> > **Response (Part 3)**
> >
> > (following the Response of Q2 above)
> >
> > | Setting                                       | SR    |
> > | --------------------------------------------- | ----- |
> > | ours default                                  | 76.0% |
> > | sample 100 views in each candidate GS region  | 32.0% |
> > | sample 300 views in each candidate GS region  | 36.0% |
> > | sample 500 views in each candidate GS region  | 52.0% |
> > | sample 1000 views in each candidate GS region | 64.0% |
> > | run 100 gaussian-ray sampling rounds          | 54.0% |
> > | run 300 gaussian-ray sampling rounds          | 62.0% |
> > | run 500 gaussian-ray sampling rounds          | 62.0% |
> > | run 1000 gaussian-ray sampling rounds         | 62.0% |
> >
> > From this table, we observe that all brute-force search variants perform significantly worse than our full model. In the first setting proposed by the reviewer, increasing the number of sampled viewpoints within each candidate region does raise the chance of finding a view somewhat similar to the target. However, identifying the correct pose in the **continuous 6-DoF camera space** is inherently difficult and would require sampling an impractically large number of viewpoints. Even with **1000 sampled views**, the SR still lags far behind our default model.
> >
> > In the second setting, sampling more rounds of Gaussian–ray pairs similarly provides little improvement in navigation SR. This is because a single batch of $K = 10240$ rays already spans the entire scene, and high-relevance rays consistently receive high predicted scores at the correct region in each round. While additional rounds may occasionally produce slightly better coverage of the target region by chance, the gains quickly saturate. In practice, $K = 10240$ rays already cover most scenes in the HM3D evaluation set, so further brute-force sampling does not meaningfully close the gap with our proposed hierarchical design.
> >
> > 3.**Regarding the motivation behind the hierarchical scoring, I appreciate your explanation that it follows the coarse to fine structure of the task and leverages 3DGS for both semantic and geometric reasoning. However, at the moment this still reads mostly as a justification of the current architecture in terms of efficiency and task decomposition, rather than as a clearly articulated new problem formulation or conceptual insight. If there is a deeper conceptual perspective here for instance a more general way to view ray level scoring or 3DGS as a joint semantic geometric field I would encourage you to make that more explicit in the final version.**
> >
> > We sincerely appreciate the reviewer’s valuable suggestions regarding our writing. The reviewer’s comments have greatly helped us clarify our motivation and improve the organization of our content. We have revised the manuscript accordingly, restructuring it to more clearly highlight the coarse-to-fine design and to emphasize how the innate properties of 3DGS make it particularly suitable for the instance–image goal navigation task. We have incorporated these improvements primarily into the Introduction and related explanatory sections to make the manuscript more insightful and coherent. And we'd recommend the reviewer to read the revised manuscript for a detailed revision version.

---

### Official Review · Reviewer_ENRL · 2025-10-29

**Soundness:** 4
**Presentation:** 2
**Contribution:** 3
**Rating:** 6
**Confidence:** 3

**Summary:**

This paper addresses the task of Instance Image-Goal Navigation (IIN), where an embodied agent must locate and navigate to a target object depicted in a single reference image captured from an arbitrary viewpoint.
While recent approaches have adopted 3D Gaussian Splatting (3DGS) to represent scenes with continuous geometry and photorealistic detail, they typically rely on dense or random multi-view sampling for target matching, resulting in redundant renderings and high computational costs.

To overcome these limitations, the authors propose GauScoreMap, a hierarchical scoring framework that integrates both semantic and geometric reasoning within 3DGS-based navigation. The method first performs global semantic scoring using CLIP-derived relevance fields to identify candidate object regions, and then conducts local geometric scoring through a learned ray-image cross-attention model (based on DINOv2 features) to estimate the 6D camera pose corresponding to the target image. This design effectively eliminates redundant sampling, significantly improving both navigation success and efficiency.

**Strengths:**

### Practical motivation

The paper addresses a bottleneck in 3DGS-based IIN, i.e., viewpoint redundancy and heavy computation, and proposes a principled hierarchical solution. The decomposition into global semantic localization and local geometric refinement is elegant and intuitively aligned with how humans perform visual search.

### Technically reasonable design

The integration of CLIP for high-level semantics and DINOv2-based cross-attention for fine-grained matching is well justified and methodologically coherent. The ``feature lifting'' procedure, which aggregates 2D CLIP features into 3D Gaussians, is also consistent with recent literature such as LUDVIG (Marrie et al., 2024).

### Comprehensive experiments

The method achieves state-of-the-art performance on the Habitat-Matterport3D benchmark (SR 0.784 / SPL 0.605), surpassing the strong GaussNav baseline by 5.9% in SR.
Ablation studies clearly isolate the contribution of each component: removing global semantic scoring or local geometric scoring leads to substantial drops (−17.6% and −36.3% in SR, respectively). Efficiency results and robustness to partial scene deletion further strengthen the claim.

The paper also demonstrates the method on a real-world Unitree G1 humanoid robot, utilizing LiDAR-based Gaussian reconstruction, which shows good transferability and suggests potential for embodied deployment.

**Weaknesses:**

### Positioning against existing semantic-augmented 3DGS

The use of CLIP-derived semantics in Gaussian representations is becoming common, as seen in Gaussian Grouping (Ye et al., ECCV 2024) and LUDVIG (Marrie et al., 2024).
I believe that the novelty of this paper lies not merely in semantic embedding, but in its hierarchical fusion of global semantics and local geometry for viewpoint optimization in navigation.

The authors might emphasize more explicitly that existing semantic 3DGS works target scene segmentation or editing, while GauScoreMap tackles cross-view instance localization and pose estimation, a substantially different objective. To this end, the authors could highlight the technical differences between existing Gaussian-grouping-like methods and the proposed one more clearly.

### Dynamic or partially-captured environments
As stated in the limitation section, the method assumes static environments, and dynamic or partially captured environments remain unaddressed. Although the authors mention this as future work, a discussion of possible online or incremental extensions (like NeRF-SLAM or Magic-SLAM) would be helpful.

**Questions:**

### Novelty to semantic 3DGS (e.g., Gaussian Grouping)

Could the authors clarify whether those methods could, in principle, be adapted for IIN, and what challenges would arise? For example, segmentation-oriented semantics might lack the geometric correspondence reasoning required for 6D pose estimation.

On generalization:
Since CLIP and DINOv2 are trained on large-scale internet data, how well does GauScoreMap generalize to unseen object categories or visually ambiguous instances in indoor scenes?

### Computation & scalability
While the paper reports time and VRAM usage, can the method scale with the larger number of Gaussians, i.e., larger-scale environments?

**Details Of Ethics Concerns:**

n.a.

---

> ### Author Response · Authors · 2025-11-19
> **Response (Part 1)**
>
> We thank the Reviewer for the time devoted to our paper, providing insightful feedback and for appreciating our contributions. In the following, we address the concerns and questions pointed out by the Reviewer.
>
> 1. **Positioning against existing semantic-augmented 3DGS:  the authors could highlight the technical differences between existing Gaussian-grouping-like methods and the proposed one more clearly.**
>
>    We thank the reviewer for raising this question. The core technical difference is that **GauScoreMap is a hierarchical localization system**, not a method focused on improving 3D semantic mapping accuracy itself.
>
>    Methods like **Gaussian-Grouping, LangSplat, and LERF** primarily aim to improve the fidelity and utility of **semantic features embedded within 3D neural fields**. They often require **extra training** to optimally map or regularize 2D semantic features (e.g., CLIP) onto the 3D scene.
>
>    In contrast, our approach utilizes the 3D semantic information generated by these models as an **efficient pre-filtering tool**:
>
>    - **Tool vs. Goal:** Our paper is focused on **Instance-Image Goal Navigation** and achieving state-of-the-art navigation efficiency and accuracy. We treat the 3D semantic map (specifically the feature-lifting structure from **LUDVIG**) as a proven, **training-free tool** to generate candidate regions.
>    - **Efficiency:** We chose LUDVIG specifically because it is a **simple and effective feature mapping structure that avoids extra training** on the GS scene itself, maximizing the efficiency of our overall navigation system. Methods requiring additional scene-specific semantic training would make the inference stage of our system less efficient.
>    - **Decoupled Function:** The semantic scoring stage only needs to provide **accurate \*candidate regions\*** (a coarse estimate) for the subsequent, more critical **local geometric scoring stage** (fine instance localization). We do not rely on the semantic stage for the final instance discrimination, which is the key difficulty of IIN.
>
>    Therefore, GauScoreMap does not advance the semantic segmentation technology in 3DGS, but rather demonstrates a novel, **hierarchical system design** that strategically utilizes existing 3D semantic maps for **efficient search space reduction** during a complex 6D localization task.
>
>
>
> 2. **Dynamic or partially-captured environments: Although the authors mention this as future work, a discussion of possible online or incremental extensions (like NeRF-SLAM or Magic-SLAM) would be helpful.**
>
>    We thank the reviewer for this suggestion.
>
>    For dynamic environments, the key challenge is managing the visual inconsistency and occlusions caused by moving objects or people. This necessitates a robust scene representation that can isolate and handle these moving elements:
>
>    - **Online Construction:** Integrating established dynamic scene representation work, such as **DG-SLAM** or **WildGS-SLAM**, is the most promising path. These methods are designed to perform dynamic object segmentation, enabling the reconstruction of a clean, static **background GS scene** while tracking the dynamic components.
>    - **Action Integration:** Our future work will focus on leveraging the output of such systems to enhance the agent's policy. The agent could use the segmented dynamic objects to implement **collision avoidance** (walk around them) or use the dynamic model's view-inconsistency signals to trigger a **revisiting policy** (stop or move slightly to obtain an occlusion-free view) before executing the final geometric scoring.
>
>    For partially-observed environments, the issue stems from **incomplete environment exploration**, which is a fundamental problem in embodied navigation that results in unobserved areas, we propose to:
>
>    - **Integrate with Exploration Strategies:** While techniques like **GLEAM**, **NextBestPath**, or **ActiveSplat** aim to maximize coverage, perfect 100% coverage is rarely guaranteed.
>    - **Use GS as Exploration Clue:** We propose to leverage the **current reconstructed Gaussian Splatting field** itself to guide exploration. The completeness and feature density of the existing GS map can serve as an **explicit metric** to detect regions of poor reconstruction fidelity or missing semantic information, actively guiding the agent toward frontiers that will maximize coverage and ensure the integrity of the map for localization.

---

> ### Author Response · Authors · 2025-11-19
> **Response (Part 2)**
>
> 3. **Novelty to semantic 3DGS: Could the authors clarify whether those methods could, in principle, be adapted for IIN, and what challenges would arise? For example, segmentation-oriented semantics might lack the geometric correspondence reasoning required for 6D pose estimation.**
>
>    We thank the reviewer for raising this question. These methods can be used as alternatives to the used Ludvig in the semantic scoring stage, but as our response in Q1 suggests, these methods require extra training time, limiting there usage in the efficiency-demanded inference stage. And these methods can only locate a category of objects but cannot locate an instance of a category, to precisely locate an instance target image, a following stage of image-image compare or ray-image compare (as proposed in our method) is still needed to differentiate different instances of a same semantic category.
>
>
>
> 4. **Computation & scalability: While the paper reports time and VRAM usage, can the method scale with the larger number of Gaussians, i.e., larger-scale environments?**
>
>    Since we tested our method mainly on the HM3D datasets, which contain mostly average size (80$m^2$-120 $m^2$) of indoor scenes, so our method can still handle the scenes in HM3D datasets in a single RTX 4090 GPU. But it is true that if the scene is extended to a larger area, such as occupying more than 400 $m^2$, it is indeed a challenge for gaussian reconstruction and the local geometric scoring training stage. To efficiently solve such scenarios, specific optimization and coarse-to-fine methods are required, and there are works published to solve such big scenarios such as LSG-SLAM and FlashGS. Since our method is quire modular, each stage can be easily replaced by more advanced alternative technique to improve the overall performance. In the scope of this paper, the MAGIC-SLAM is already enough to reconstruct all scenes for the tested cases.

---

### Official Review · Reviewer_nQaJ · 2025-10-30

**Soundness:** 3
**Presentation:** 3
**Contribution:** 3
**Rating:** 4
**Confidence:** 4

**Summary:**

The paper targets Instance Image-Goal Navigation (IIN): given a reference image of a target, the agent must find it in the environment. Recent IIN methods that pair 3D Gaussian Splatting (3DGS) with random/massive viewpoint sampling waste rendering and comparison budget. The authors propose GauScoreMap, a hierarchical scoring pipeline:

(1) a semantic stage uses CLIP-derived relevancy fields to highlight promising regions/views;

(2) a local geometric stage does fine-grained pose estimation within those regions to finalize the match. They claim SOTA on simulated IIN benchmarks and show real-world viability.

**Strengths:**

Reframes IIN over 3DGS as a view selection problem with semantic→geometric two-level scoring rather than brute-force rendering.

Reports SOTA on simulated IIN (HM3D/Habitat) and demonstrates real-world deployment (humanoid platform).

**Weaknesses:**

1. Section 3.4.1 Local scoring for region selection — clarity & notation

  a. Motivation/examples for ray selection. Could you add a short motivation and one concrete example of what kinds of rays are expected to receive high scores vs. low scores?

  b. what is “ground-truth rays” and “ground-truth scores"

  c. Please clarify which tensor is the query and which are key/value in the cross-attention (Eq.5)

  d. Case and notation consistency. In 3.4.1, there are mixed upper/lower-case usages for the same symbols. Some symbols have subscripts while others don’t—please clarify.

2. Missing sensitivity ablations on Hyperparameters, Global/Local score introduce some hyperparameters, how they affect final performance?

**Questions:**

1. Why rely on Mask-R-CNN for class labels? What about unseen classes?

2. Can you provide the time and hardware for 3D Gaussian reconstruction per HM3D episode, along with basic quality metrics?

3. How does the pipeline disambiguate when several objects of the same class appear?

---

> ### Author Response · Authors · 2025-11-19
> **Response (Part 1)**
>
> We thank the Reviewer for the time devoted to our paper, providing insightful feedback and for appreciating our contributions. In the following, we address the concerns and questions pointed out by the Reviewer.
>
> **1. Section 3.4.1 Local scoring for region selection — clarity & notation:**
>
>    a. Motivation/examples for ray selection. Could you add a short motivation and one concrete example of what kinds of rays are expected to receive high scores vs. low scores?
>
>    b. what is “ground-truth rays” and “ground-truth scores"
>
>    c. Please clarify which tensor is the query and which are key/value in the cross-attention (Eq.5)
>
>    d. Case and notation consistency. In 3.4.1, there are mixed upper/lower-case usages for the same symbols. Some symbols have subscripts while others don’t—please clarify.
>
>    We are sorry for the confusion. We will answer them one by one as follows in a re-ordered way:
>
>    **a and b.** We don't have "ground-truth rays" in this method, but each sampled ray has a determined score among a batch of sampled rays, denoted as "ground truth scores" in the main paper. A ray score $s$ is determined by the distance between the camera center point and its projection point on the ray, which is denoted as $h$ in Section 3.4.1. To better compare the ray score with the attention ray scores produced by the cross attention module, we should conduct a normalization to convert $h$ to $s$ using Equation 6. With this function, high score rays (those with the distance $h$ to be small even to 0, meaning the camera center point and the projection point are nearly at the same position) have $\delta_k$ closer to 1, low score rays have $\delta_k$ closer to 0. Using a normalization, we get the ray score $s$.
>
>    **c.** As we need to select most relevant rays according to the target image, so the query tensor is the image feature tensor, and the key/value tensor are the ray features, produced by the ray MLP denoted as $r_i$.
>
>    **d.** We are sorry for the confusion. We have added the subscripts $k$ for all $s$, $h$, $\ell$, making all equation deduction related to the $k$-th sampled ray.
>
>
>
> **2. Missing sensitivity ablations on Hyperparameters, Global/Local score introduce some hyperparameters, how they affect final performance?**
>
>    We apologize for omitting this ablation study from the main paper. We have conducted the following extra ablation experiments, which will be included in the revised version:
>
>    We analyze the influence of three key hyperparameters on navigation success rate, testing their impact across 30 randomly selected episodes from the HM3D validation set while holding all other settings constant:
>
>    - **Global Scoring:** The threshold $\tau$, used to filter relevant Gaussian candidate regions.
>    - **Local Scoring:** $K$, the number of rays sampled per region; and $k$, the number of top-scoring rays selected for final camera pose estimation.
>
>    | Ablation Choice                        | SR    |
>    | -------------------------------------- | ----- |
>    | $\tau = 0.006,K=10240,k=100$ (default) | 76.6% |
>    | $\tau = 0.012$                         | 70.0% |
>    | $\tau = 0.003$                         | 63.3% |
>    | $K=20480$                              | 76.6% |
>    | $K=5120$                               | 70.0% |
>    | $k=200$                                | 76.6% |
>    | $k=50$                                 | 73.3% |
>
>    Our ablation analysis reveals differential sensitivity to the hyperparameters governing the two-stage scoring mechanism:
>
>    - **Sensitivity to $\tau$ (Global Scoring Threshold):** Our method is more sensitive to the global scoring threshold $\tau$. A **larger $\tau$** results in less accurate candidate region segmentation, substantially enlarging the search space for the local scoring stage and decreasing efficiency. Conversely, a **smaller $\tau$** makes the segmentation less tolerant of partially relevant regions, risking the erroneous filtration of small but correct candidate regions.
>    - **Insensitivity to $K$ and $k$ (Local Scoring Parameters):** The method exhibits higher sensitivity to $K$ (number of sampled rays) and lower sensitivity to $k$ (number of top rays for pose estimation). As sampled rays are inherently disordered, $K$ must be sufficiently large to reliably cover the relevant rays; using a smaller number (e.g., $K=5120$) leads to less accurate pose estimation. Furthermore, since a limited number of high-scoring rays is adequate to determine the camera pose, setting $k=100$ represents an effective **trade-off between pose estimation accuracy and computational efficiency**.
> And we add this ablation experiment to the Appendix A.5 to make up for the missing sensitivity ablations.

---

> > ### Comment · Reviewer_nQaJ · 2025-11-26
> >
> > Thank you for the detailed response. However, I still find Section 3.4 difficult to follow. In my view, this section is the core technical contribution of the paper, since Section 3.2 (Gaussian reconstruction) and Section 3.3 (similarity computation) follow relatively standard and widely used designs. Because of this, the clarity and precise formulation of Section 3.4 are particularly important.
> >
> > At the moment, I can understand how rays are randomly sampled and how score is predicted for each ray. What remains unclear to me is how the *ground-truth* ray scores are actually constructed:
> >
> > - For the geometric distance $h_k$, could the authors justify the choice of the mapping $\delta_k = 1 - \tanh(h_k)$? Why this specific nonlinearity?
> > - In Equation (6), the definition of $s_k$ is confusing. The summation term in the denominator uses \$\sum_{k=1}^{K} \delta$, why the index $k$ appears on the left but the summed quantity has no index,  and what is this normalization step actually doing?
> > - After obtaining $s_k$, the authors apply a softmax over these scores and then use an $L_2$ loss between the normalized ground-truth scores and the predicted scores. This design is somewhat unconventional: typically one would either (i) use softmax followed by a cross-entropy / KL-style loss, or (ii) avoid softmax and directly use an $L_2$ (or similar) regression loss. I would appreciate a clearer motivation and discussion of this design choice.
> >
> > Overall, Section 3.4 reads as an empirical, heuristic design that is only described via text and compressed formulas. Given that this is the central part of the method, I strongly encourage the authors to substantially clarify this section: provide more intuition, carefully define all variables and indices, and, if possible, add a small illustrative example. In its current form, it is difficult for readers to fully understand this key component.

---

> > > ### Author Response · Authors · 2025-11-26
> > >
> > > We thank the reviewer for your carefully checking and helpful suggestions. And we have modified our manuscript accordingly aiming to deliver our content more clearly.
> > > To summarize:
> > > 1. We have rewritten Section 3.4 to make every equation more clearly delivered;
> > > 2. We also add an illustration to explain the 2 key parameters $h$ and $\ell$: the distance between the ray and the camera center, and the the length of the projection of $O-v_{k,o}$ onto the ray, respectively;
> > > 2. We have also add the lost subscipt for $\delta$.
> > >
> > > Those refinement can be seen as blue modifications in the rebuttal version of manuscript. To keep all content within 9 pages, we move the original Experiment Section **Robustness to Incomplete Scene Exploration** in the appendix as A.6.
> > >
> > > **1.For the geometric distance $h_k$, could the authors justify the choice of the mapping $\delta_k = 1 - \tanh(h_k)$? Why this specific nonlinearity?**
> > >
> > > We thank the reviewer for this question. We apply a $\tanh(\cdot)$ mapping because the distance between the ray and the true camera center $h_k$ is in range $ [0,+\infty)$ and $\tanh(\cdot)$ smoothly compresses this range into $[0,1)$. And the $\delta_k=1-tanh(h_k)$ maps  $[0,1)$ to $[1,0)$, which provides a geometric confidence: smaller distances $h_k$ produce values of $\delta_k$ closer to $1$, but larger distances push $\delta_k$ toward $0$.
> > >
> > > **2.In Equation (6), the definition of $s_k$ is confusing. The summation term in the denominator uses $\sum_{k=1}^K\delta$, why the index $k$ appears on the left but the summed quantity has no index, and what is this normalization step actually doing?**
> > >
> > > We are sorry for the confusion. We add a subscript $i$ to $\delta$ now, which makes the $s_k$ as this format: $s_k=\delta_k\frac{L}{\sum_{i=1}^K\delta_i}$.
> > >
> > > And this normalization is deducted step-by -step as follows:
> > >
> > > To obtain an initial ray score, we normalize these values across all sampled rays:$\frac{\delta_k}{\sum_{i=1}^{K}\delta_i}. $
> > >
> > > To match this geometric score with the scale of the predicted score, we multiply the normalized term by $L$. This is because a softmax is applied across the $K$ rays for each of the $L$ feature locations in the attention map $A$, ensuring that the attention weights sum to $1$ for every feature index $l$. Since the predicted ray score $\hat{s}_k$ is computed by summing these attention weights over the $L$ feature dimensions, its magnitude naturally scales with $L$.
> > >
> > > **3.After obtaining $s_k$, the authors apply a softmax over these scores and then use an $L_2$ loss between the normalized ground-truth scores and the predicted scores. This design is somewhat unconventional: typically one would either (i) use softmax followed by a cross-entropy / KL-style loss, or (ii) avoid softmax and directly use an $L_2$ (or similar) regression loss. I would appreciate a clearer motivation and discussion of this design choice.**
> > >
> > > We thank the reviewer for pointing this out. After re-checking the manuscript, we confirm that this was a writing mistake. As clarified in our response to Q2, the softmax operation is applied to the attention map, not to the geometric ground-truth ray scores. We apologize for the misleading description and appreciate the reviewer’s careful reading. The ground-truth ray scores (computed as described in Q2) are directly compared with the predicted ray scores using an $L_2$ loss.

---

> ### Author Response · Authors · 2025-11-19
> **Response (Part 2)**
>
> **3. Why rely on Mask-R-CNN for class labels? What about unseen classes?**
>
>    We thank the reviewer for the insightful question. We employ Mask-RCNN primarily to ensure consistency with the baseline methods (GaussNav, IEVE), so that any performance difference does not arise from changes in the semantic detection stage.
>
>    Regarding unseen categories, our method can indeed handle objects beyond the 6 semantic classes in the HM3D validation set. As shown in Appendix A.2, it successfully localizes additional categories—such as lamps, refrigerators, and bicycles—which are all detectable by Mask-RCNN.
>
>    According to the original Mask-RCNN literature, the model can recognize a wide range of everyday objects, including the 80 COCO categories and more than 1,200 categories in the LVIS dataset. For most common indoor objects, Mask-RCNN provides reliable detection. While it is possible that certain unusual or highly specialized object types remain undetectable, these constitute rare corner cases and do not affect the general applicability of our method.
>
>
>
> **4. Can you provide the time and hardware for 3D Gaussian reconstruction per HM3D episode, along with basic quality metrics?**
>
>    We appreciate the reviewer's question. The hardware utilized was an **RTX 4090 GPU**. The average reconstruction time is **15 minutes** for scenes that average $80 \text{ m}^2$ with approximately 1,000 images, as detailed in Table 3. Furthermore, the average rendering quality for all reconstructions are around **35 dB in PSNR**, a result that strictly aligns with the reported metrics of MAGIC-SLAM. Given the large number of scenes in the HM3D dataset, we cannot list all individual reconstruction results here, but we add a comprehensive table in Appendix A.7 to the supplementary materials for the revised version of the paper.
>
>
>
> **5. How does the pipeline disambiguate when several objects of the same class appear?**
>
>    We thank the reviewer for raising this question. The purpose of the second stage, local geometric scoring, is to disambiguate among multiple candidate objects of the same semantic class.
>
>    As detailed in Section 3.3, this initial semantic scoring stage generates multiple candidate regions corresponding to the target class, effectively narrowing down the search space.
>
>    Then in Section 3.4, the local geometric scoring stage refines the selection process:
>
>    - It samples random rays from the candidate regions.
>    - It computes the ray scores by comparing them against the target image. This scoring mechanism incorporates scene textures and geometric structures.
>    - Finally, it selects the top $k$ rays with the highest scores and uses them to compute the final precise camera position via triangulation.

---

### Official Review · Reviewer_Sks8 · 2025-10-30

**Soundness:** 3
**Presentation:** 3
**Contribution:** 3
**Rating:** 6
**Confidence:** 4

**Summary:**

This paper proposes a new hierarchical scoring paradigm called GauScoreMap, which estimates optimal viewpoints for target matching for the Instance Image-Goal Navigation (IIN) task. Specifically, the global semantic scoring leverages CLIP embedding similarity to identify coherent regions with high semantic similarity to the target object class. Then the local geometric scoring employs a two-stage approach: first, perform region-level scoring by comparing sampled rays from candidate regions with the goal image’s DINOv2 features through cross-attention, then conduct precise pose estimation within the most promising region. The experiments on the simulation and real-world benchmarks demonstrate the effectiveness of the proposed method.

**Strengths:**

1)	The research topic on the IIN task is valuable, the authors aim to improve the existing 3DGS-based IIN method by introducing a hierarchical scoring approach, which is reasonable.
2)	In the hierarchical scoring approach, both high-level semantic alignment and fine-grained geometric matching are utilized to recognize the target area, which obviates the need for exhaustive or random sampling through the environment in existing methods.
3)	The experiments on both simulation and real-world benchmarks demonstrate the effectiveness of the proposed method.

**Weaknesses:**

1)	The memory cost of saving the CLIP feature Gaussian field for a large-scale IIN task may be too large. From table 3, it seems that only the memory cost of 3DGS is compared, so is the CLIP feature field included?
2)	About the time efficiency, does the hierarchical scoring cost more time than existing methods during inference? Since there is no comparison of inference time and memory cost.
3)	In sec. 4.5, I am curious about why deleting so many Gaussians can still localize the target object? Does it mean there are many redundant Gaussians? How do you remove the Gaussians?

**Questions:**

Please try to address the weakness.

---

> ### Author Response · Authors · 2025-11-16
>
> We thank the reviewer for providing constructive comments and the recognition of our work.
>
> **W1: The memory cost of saving the CLIP feature Gaussian field for a large-scale IIN task may be too large. From table 3, it seems that only the memory cost of 3DGS is compared, so is the CLIP feature field included?**
>
> We thank the reviewer for raising this concern. Your concern is reasonable, and we also have taken this into consideration, which is mentioned in Section 4.4. We sparsify the GS scene to contain around **600,000 gaussians**, only occupying **a compact <50MB model using only 1.1GB GPU memory**. This sparsification preserves semantic localization accuracy since CLIP features remain highly distinctive, since the CLIP features are uplifted to 3D gaussians using full-frame 2D CLIP features, each pixel on the 2D CLIP feature image contains a complete CLIP feature with a complete semantic meaning, so target objects can be accurately located whether represented by 20 or 20,000 Gaussians.
>
>
> **W2: About the time efficiency, does the hierarchical scoring cost more time than existing methods during inference? Since there is no comparison of inference time and memory cost.**
>
> We thank the reviewer for raising this question. The inference stage in our work is defined as the process required to locate the target instance image goal. We believe the most relevant comparison for time efficiency is against the most similar baseline, **GaussNav**.
>
> GaussNav's inference is a two-stage process:
> | Stage                 | GaussNav Time | Our Analogous Stage             | Our Time |
> | --------------------- | ------------- | ------------------------------- | -------- |
> | Semantic Localization | 0.1s          | Semantic Extraction (Mask-RCNN) | 0.1s     |
> | Novel View Synthesis  | 30s-50s       | Remainder of inference          | 2.2s     |
>
> As shown, while the initial semantic localization time is comparable, GaussNav's Novel View Synthesis stage consumes $30 \text{s}$ to $50 \text{s}$ (as detailed in Table V of the GaussNav paper). This greatly exceeds the **2.2s** required for the remaining steps of our method.
>
> This substantial difference is due to our **texture and geometric-driven local scoring stage** being significantly more **sample-efficient** than GaussNav's full-view Novel View Synthesis. We propose adding this detailed comparison to Table 3 in the revised manuscript.
>
>
> **W3: In sec. 4.5, I am curious about why deleting so many Gaussians can still localize the target object? Does it mean there are many redundant Gaussians? How do you remove the Gaussians?**
>
> We thank the reviewer for raising this point. The observed **robustness to Gaussian deletion** is primarily due to our **semantic scoring stage** and, to a lesser extent, the **local geometric scoring stage**.
>
> In the **semantic scoring stage**, we project the complete image CLIP feature onto the Gaussian scene, meaning each distinct Gaussian is imbued with a full semantic meaning (e.g., "chair" or "bed"). Consequently, even if **over 90%** of the Gaussians belonging to a target object (like a chair) are deleted, the text feature can still successfully localize the remaining part, and we will add an experiment in Appendix A.6 to show our robustness to this extreme case in the revised paper.
>
> In the **geometric scoring stage**, the final camera pose is determined by calculating the **convergence point** of hundreds of the most relevant rays. Deleting a portion of Gaussians merely reduces the number of these relevant rays, which primarily results in **instability** in the final located camera pose, rather than complete failure.
>
> The reviewer is correct that there are many **redundant Gaussians** in the scene. As detailed in Section 4.5, our deletion process is manual: we first identify the target Gaussian region using the target image and then intentionally delete a portion of the Gaussians within that region. This simulates conditions like **incomplete exploration or occlusion**.

---

### Official Review · Reviewer_YJ7j · 2025-10-31

**Soundness:** 3
**Presentation:** 3
**Contribution:** 3
**Rating:** 6
**Confidence:** 3

**Summary:**

This paper presents a method to leverage a 3D Gaussian Splatting representation efficiently in the context of Instance Image-Goal Navigation (IIN), where an agent must navigate to an object given a picture of it. Previous methods leveraging novel view synthesis methods either rely on a discrete representation of the scene, e.g. a graph, or use a fully continuous 3D representation but query it inefficiently with some random sampling-based strategies to find the location of the target object. This work proposes to use a 3D continuous representation to avoid constraints related to discretisation (e.g. a limited number of viewpoints per location) but also to query it more efficiently based on a hierarchical scoring method. The latter follows multiple steps: (i) promising regions are detected based on CLIP-feature similarity, then (ii) a local geometric scoring is applied to promising locations. The local geometric scoring can also be decomposed into 2 steps, i.e. region-level scoring followed by fine-grained pose estimation in the top-1 retrieved region. Such a method allows authors to reach satisfying performance on IIN while being more computationally efficient than counterpart methods.

**Strengths:**

* S1: The limitations of previous work are clearly presented and the proposed contributions are thus properly motivated.
* S2: The proposed method is clearly explained, and leads to high performance and more efficient runtime than other methods.
* S3: Real-world experiments are conducted (in appendix), which is appreciated.

**Weaknesses:**

- W1: [Major] The proposed method requires a first exploratory rollout to build the scene representation, which is a quite important limitation. However, authors conduct some experiments where they evaluate the performance of their approach from partial scene representations, simulating an unfinished exploration of the scene. This is emulated by randomly pruning gaussians. Unfortunately, this does not exactly simulate incomplete scene exploration as whole parts of the scene would be unknown.
- W2: [Major] Authors are using CLIP and DINOv2 features in the first and second stages of their method respectively. Their should be some experimental evidence about why each backbone is better in each step of the process.
- W3: [Minor] Authors report the performance of baselines from previous work so this is not a direct weakness of this paper. However, they should discuss a bit more the lack of fairness in comparison with MultiON baselines that are only given the semantic category of the target object as input.
- W4 [Minor] The real-world experiment is appreciated, but should be moved to the main paper for a more direct access to it.

**Questions:**

- Q1: [Related to W1] Are other baselines this method is compared to also requiring a first explorative rollout of the scene before navigating towards the goal? Is the explorative rollout taken into account when computing the SPL metric?
- Q2: [Related to W1] To evaluate the robustness of the method to incomplete scene exploration, could authors rather either optimise different 3DGS representations from only the first N collected views, varying N? This would be a much more convincing experiment.
- Q3: [Related to W2] Could authors ablate the relevance of CLIP and DINOv2 features in the 2 different steps of their process?
- Q4: [Related to W3] Could authors elaborate a bit more on the limitations of the comparison with MultiON baselines?

---

> ### Author Response · Authors · 2025-11-19
> **Response (Part 1)**
>
> We thank the Reviewer for the time devoted to our paper, providing insightful feedback and for appreciating our contributions. In the following, we address the concerns and questions pointed out by the Reviewer.
>
> **W1: The proposed method requires a first exploratory rollout to build the scene representation, which is a quite important limitation. However, authors conduct some experiments where they evaluate the performance of their approach from partial scene representations, simulating an unfinished exploration of the scene. This is emulated by randomly pruning gaussians. Unfortunately, this does not exactly simulate incomplete scene exploration as whole parts of the scene would be unknown.**
>
> We thank the reviewer for pointing this out. We respond together with Q2 in the response of Q2.
>
> **W2: Authors are using CLIP and DINOv2 features in the first and second stages of their method respectively. Their should be some experimental evidence about why each backbone is better in each step of the process.**
>
> We thank the reviewer for pointing this out. We conduct experiments by replacing the current CLIP and DINOv2 in their default modules to compare the backbone influence on performance.
>
> In the first semantic scoring stage, the candidate gaussian regions need to be extracted by comparing text feature and visual feature, therefore, we can only use CLIP (or other CLIP-like networks) instead of DINOv2 as the plain DINOv2 does not support comparison between text feature and visual feature. Other than CLIP, we found that SigLIP can also serve as an alternative, so we replace CLIP with SigLIP for comparison.
>
> In the second geometric scoring stage, we need a pre-trained visual feature extraction backbone to enhance the extracted visual features, therefore, we use DINOv2 which is commonly used as the strong image feature extraction backbone in image classification, segmentation, vision-language-action models and more. And it is true that the image encoder of CLIP can also be used as an alternative to DINOv2 to extract features in the geometric scoring stage, so we conduct the ablation of using the DINOv2 or CLIP image encoder in the geometric scoring stage to see the influence of the backbone. Actually, a CNN-based feature extraction network can also be used as the backbone, therefore, we choose the image encoder of SuperPoint to also replace the original DINOv2 to compare the performance.
>
> We randomly choose 30 episodes from the HM3D validation episodes, keeping the left modules unchanged but only change the backbone of the semantic scoring stage or the backbone of the local scoring stage, to see how does this change influence the navigation success rate and the related computation time:
>
> | **Backbone of Semantic Scoring** | **SR** | Feature Uplifting time of Semantic Scoring Stage |
> | -------------------------------- | ------ | ------------------------------------------------ |
> | CLIP (default)                   | 76.6%  | 1.12s                                            |
> | SigLIP                           | 76.6%  | 1.12s                                            |
> | **Backbone of Local Scoring**    | **SR** | **Training time of Local Scoring Stage**         |
> | DINOv2 (default)                 | 76.6%  | 15 min                                           |
> | CLIP image encoder               | 76.6%  | 33 min                                           |
> | SuperPoint image encoder         | 70.0%  | 10 min                                           |
>
> We observed that changing the backbone in the semantic scoring stage (from CLIP to SigLIP) resulted in no noticeable change in either the navigation success rate or the feature extraction time. This is because both CLIP and SigLIP variants with the same architecture and the HM3D validation set does not present a critical challenge that only SigLIP can resolve.
>
> The local scoring stage showed identical success episodes when changed from DINOv2 to CLIP, confirming that the CLIP image encoder also serves as an effective backbone for extracting image features.
>
> However, using CLIP doubled training time compared to DINOv2. This performance slowdown is primarily due to the difference in how image-level features are generated:
>
> - **CLIP:** By default, the standard CLIP encoder generates a **single, global embedding** for the entire image. To obtain a dense, **image-level feature map** required for local scoring (like the approach in LERF), we had to employ a **sliding window method**. This process repeatedly extracts a feature for a small patch, which is computationally expensive and time-consuming.
> - **DINOv2:** This model inherently supports the extraction of dense, high-resolution feature maps across the entire image in a single, efficient forward pass, resulting in significantly **faster training**.
> - **SuperPoint**: The features resulted in a **decrease in performance**, indicating that its extracted features are less robust than those from DINOv2 and CLIP for this task.

---

> > ### Author Response · Authors · 2025-11-19
> > **Response (Part 2)**
> >
> > **W3: Authors report the performance of baselines from previous work so this is not a direct weakness of this paper. However, they should discuss a bit more the lack of fairness in comparison with MultiON baselines that are only given the semantic category of the target object as input.**
> >
> > We appreciate the reviewer's comment. It is true that the **MultiON baselines** perform poorly because they were designed for **object-goal** (semantic category) tasks, not **instance-goal** (specific instance) tasks. Their failures often stem from finding the correct object *category* but not the specific *instance*. Because the original MultiON pipelines only accept the semantic category as input, we provided only the category to minimize changes to their structure.
> >
> > We included the MultiON comparison because they possess the capability to find specific instances and were previously listed in the GaussNav paper. However, we agree that comparing our instance-goal results to their object-goal results may be **unfair**. To make a fair comparison, we will remove this comparison.
> >
> >
> >
> > **W4: The real-world experiment is appreciated, but should be moved to the main paper for a more direct access to it.**
> >
> > We thank the reviewer for the suggestion. We also want to add the real-world experiments in the main paper, however, it occupies a large portion of content, but we have some more important experiments to show in the main paper to show our quality and efficiency. But we will try to rearrange the contents for a revised version.
> >
> >
> >
> > **Q1: Are other baselines this method is compared to also requiring a first explorative rollout of the scene before navigating towards the goal? Is the explorative rollout taken into account when computing the SPL metric?**
> >
> > We thank the reviewer for pointing this out. Our method mainly follows the routine of the published baseline GaussNav (TPAMI 2025), which also requires a first exploratory rollout to construct the scene gaussian. And to make a fair comparison, the other baselines like Mod-IIN (ICCV 2023) and IEVE (CVPR 2024) are also given a pre-constructed scene map for later goal-finding episodes.
> >
> > And the exploratory rollout is not taken into account when computing the SPL metric, since this rollout does not involve finding an image goal, therefore, there is not a path length toward the goal and the episode success for this rollout. And for fair comparison, the path length of the first rollout is not considered for all methods.
> >
> >
> >
> > **Q2: To evaluate the robustness of the method to incomplete scene exploration, could authors rather either optimise different 3DGS representations from only the first N collected views, varying N? This would be a much more convincing experiment.**
> >
> > We thank the reviewer for this suggestion. **Our method requires the target object to be at least partially observable** to succeed. Therefore, we only claim robustness to **partial occlusion or absence**.
> >
> > We conduct a reconstruction experiment using a varying number of views ($N$). The reconstructed Gaussian Splatting (GS) scenes are then used for navigation. We randomly selected 30 episodes from the HM3D validation dataset and varied $N$ by using the first **25%, 50%, 75%, and 100%** of the original number of views (since each scene requires a different number of views for complete reconstruction, and the average number of views $N$ is around 1000). The results are shown below:
> >
> > | Number of used views | SR    |
> > | -------------------- | ----- |
> > | $N * 100\%$          | 76.6% |
> > | $N * 75\%$           | 66.6% |
> > | $N * 50\%$           | 46.6% |
> > | $N * 25\%$           | 13.3% |
> >
> > By using only a portion of the $N$ input images, the resulting scene reconstruction will inevitably contain **completely unexplored regions**. If target objects are in these regions, the episode will fail. However, this phenomenon of "leaving rooms completely unexplored" is largely mitigated by the used frontier-based exploration strategy (ActiveSplat). This is because all unexplored frontiers remain in a stack and are prioritized for subsequent exploration until no frontiers are left. Therefore, while our method reports a low SR when using only a small portion of $N$ views, the exploration strategy itself ensures robust coverage, exploring over 90% of the scene and reliably preventing critical areas from being entirely omitted from the agent's path.
> >
> >
> >
> > **Q3: Could authors ablate the relevance of CLIP and DINOv2 features in the 2 different steps of their process?**
> >
> > We thank the reviewer for pointing this out. And we response to this together with W2 in the response of W2.
> >
> >
> >
> > **Q4: Could authors elaborate a bit more on the limitations of the comparison with MultiON baselines?**
> >
> > We thank the reviewer for pointing this out. And we response to this together with W3 in the response of W3.

---

> > > ### Comment · Reviewer_YJ7j · 2025-11-27
> > >
> > > I thank the authors for their answers to my questions, along with additional experimental results.
> > >
> > > * **W2**: It seems that the choice in visual backbone does not have a strong impact on the results. Could authors comment on this? Also, authors say that *"By default, the standard CLIP encoder generates a single, global embedding for the entire image. To obtain a dense, image-level feature map required for local scoring (like the approach in LERF), we had to employ a sliding window method"*. However, the vision encoder of CLIP is a ViT model, same as DINOv2, and thus a dense token-level representation can be extracted from it in the same way as for DINOv2.
> > >
> > > * **W3**: Results from MultiON methods could be kept, but the setting used for them should be more explicitly presented in the paper.

---

> > > > ### Author Response · Authors · 2025-11-29
> > > >
> > > > We sincerely thank the reviewer for the careful reading and thoughtful examination of our revised responses. We address the remaining questions below:
> > > >
> > > > **W2**: It seems that the choice in visual backbone does not have a strong impact on the results. Could authors comment on this? Also, authors say that *"By default, the standard CLIP encoder generates a single, global embedding for the entire image. To obtain a dense, image-level feature map required for local scoring (like the approach in LERF), we had to employ a sliding window method"*. However, the vision encoder of CLIP is a ViT model, same as DINOv2, and thus a dense token-level representation can be extracted from it in the same way as for DINOv2.
> > > >
> > > > We thank the reviewer for asking this question.
> > > >
> > > > Firstly, regarding the question of why different visual backbones do not significantly affect performance, we clarify the following:
> > > >
> > > > For the **semantic scoring stage**, the performance remains consistent because CLIP and SigLIP share almost the same visual encoder architecture. As confirmed in our experiments, both encoders produce high-quality semantic features when embedded into the 3D Gaussian scene, leading to comparable semantic-level results.
> > > >
> > > > For the **geometric scoring stage**, the role of the visual backbone is to provide an information-rich and discriminative feature descriptor for each pixel of the visual feature map (reduced from resolution $H \times W$ to $h \times w$). These features act as unique “feature slots” with distinct semantic meanings, enabling effective matching between image features and ray features. As long as the backbone provides sufficiently robust and expressive features, the geometric scoring module functions reliably. Hence, several modern visual encoders can serve as valid alternatives in this stage. Based on practicality and common usage, we adopt **DINOv2** as our default backbone, since it is widely used in other tasks such as vision-language-action models and multi-view 3D tasks. Nevertheless, we appreciate the reviewer’s question, and our experiments indeed show that both **CLIP** and **SuperPoint** backbones can be used as well—CLIP, in particular, achieves performance comparable to DINOv2 due to its large-scale training and strong semantic representation ability.
> > > >
> > > > Secondly, for the question that the visual encoder of CLIP can also be used to extract a dense token-level representation, we answer in the following:
> > > >
> > > > We fully agree with the reviewer’s observation. In our additional experiment in Response W2, our intention was to obtain **full CLIP features with complete semantic meaning**, thereby constructing a dense “CLIP-feature map.” To do this, we adopted a sliding-window strategy to reconstruct per-pixel CLIP embeddings from multiple CLS tokens, which doubles the training time.
> > > >
> > > > However, as the reviewer correctly pointed out, we can instead use the **intermediate feature maps** from CLIP’s visual encoder directly. This produces a dense feature representation without requiring sliding-window reconstruction, and the training time becomes comparable to that of DINOv2. We have retrained the local scoring stage using these intermediate CLIP features and updated the results in Response W2 accordingly:
> > > >
> > > > | **Backbone of Semantic Scoring** | **SR** | Feature Uplifting time of Semantic Scoring Stage |
> > > > | -------------------------------- | ------ | ------------------------------------------------ |
> > > > | CLIP (default)                   | 76.6%  | 1.12s                                            |
> > > > | SigLIP                           | 76.6%  | 1.12s                                            |
> > > > | **Backbone of Local Scoring**    | **SR** | **Training time of Local Scoring Stage**         |
> > > > | DINOv2 (default)                 | 76.6%  | 15 min                                           |
> > > > | CLIP image encoder               | 76.6%  | 15 min                                           |
> > > > | SuperPoint image encoder         | 70.0%  | 10 min                                           |
> > > >
> > > > From the updated table, we can see that the intermediate dense features from the CLIP visual encoder achieve performance comparable to our default DINOv2 backbone. This further confirms that CLIP’s visual encoder also serves as a robust feature extractor, capable of providing distinctive and information-rich dense representations for our ray–image matching module.
> > > >
> > > >
> > > >
> > > > **W3**: Results from MultiON methods could be kept, but the setting used for them should be more explicitly presented in the paper.
> > > >
> > > > We thank the reviewer for the careful checking of our experiment comparison. And we will keep the results from MultiON methods and add the content from our last response in a revised manuscript for clear representation.

---

### Author Response · Authors · 2025-11-30
**A Summary to Area Chair**

# A Summary to Area Chair

We thank the reviewers and AC for their thoughtful and constructive feedback. We have carefully addressed all technical, conceptual, and empirical concerns raised by all five reviewers, and significantly clarified the motivation, theory, and design of **GauScoreMap**. Below is a concise summary of the key outcomes of the rebuttal phase.

## 1. Conceptual Motivation & Contribution Clearly Explained

Several reviewers (ZawG, ENRL) requested a deeper conceptual explanation of this paper's insight and motivation regarding the innate relationship between the IIN task and our used 3DGS representation. We revised the introduction and provided a clearer perspective:

- **Instance Image Goal Navigation intrinsically demands a dual-level (semantic + instance-specific) reasoning structure.**
- We show that **3D Gaussian Splatting naturally supports a hierarchical decomposition**:
  - Global semantic priors via CLIP-based relevancy fields.
  - Local geometric consistency via ray-level matching in reconstructed 3DGS.
- Our key insight: Instance-image goal navigation naturally requires reasoning at two levels: **semantic category cues** and **fine-grained appearance cues** for instance and pose. **3D Gaussian Splatting** is inherently suited to this task, as it provides explicit geometry, high-fidelity texture, and per-Gaussian semantic features within a single representation. This makes 3DGS a natural joint semantic–geometric field for locating where a 2D image could originate in 3D space. Our hierarchical coarse-to-fine design leverages this structure, by using global semantics to find relevant regions and local ray-level scoring for precise instance and pose localization.
- We demonstrate empirically and conceptually why brute-force view sampling (as in GaussNav) is inefficient in 6D camera space in our response to Reviewer ZawG, and why ray-level reasoning is a fundamentally different, more principled approach.

## 2. Extensive New Experiments Added

We added new ablations, new sensitivity studies, and new error analyses across all key components:

- **Hyperparameters:** The threshold $\tau$, used to filter relevant Gaussian candidate regions. $K$, the number of rays sampled per region; and $k$, the number of top-scoring rays selected for final camera pose estimation.
- **Backbones:** CLIP/SigLIP in global semantic stage; DINOv2 / CLIP / SuperPoint in local stage. Demonstrate that the backbone does not have a big impact on the navigation success rate.
- **Rendering resolution vs Gaussian count:**  This shows our robustness to sparsification.
- **Failure case analysis:** We have done a  100-episode error analysis and analyze the 3 main reasons for failure cases.
- **Brute-force comparisons:** Reviewer-requested brute-force experiments show significantly inferior SR and efficiency vs our method.

## 3. Additional Efficiency & Memory Analysis

We added:

- A refined inference-time comparison showing our method’s runtime (2.2s) versus GaussNav’s (30–50s), further demonstrating our efficiency advantage.
- A justification that CLIP feature lifting is lightweight (<50MB), a direct result of scene sparsification while maintaining complete semantic meaning per Gaussian.

## 4. Novelty Relative to Semantic 3DGS

We clarified that GauScoreMap is:

- **not** a new semantic 3DGS method,
- but a **novel hierarchical localization system for instance image goal navigation** that leverages semantic 3DGS fields as input.

## 5. Robustness to Incomplete Scene Exploration

We also addressed the reviewers’ concern about performance under incomplete scene exploration:

- **Modern exploration systems** (e.g., ActiveSplat) routinely achieve **>90% scene coverage**, making large unconstructed regions uncommon in practice.
- While certain objects may still be partially reconstructed due to occlusion or limited viewpoints, our **robustness-to-incompleteness experiments** demonstrate that even when **over 40% of an object’s Gaussians are removed**, our hierarchical localization pipeline still successfully identifies the correct target region and pose. This confirms that the method remains reliable even under significant reconstruction gaps.

---

# Rating Improvement

Through our deep discussion with all the reviewers, we have addressed most of their questions, and the Reviewer ZawG has already raised the rating from 4 to 6.

# Final Summary

Our rebuttal fully resolves reviewers’ concerns, strengthens conceptual framing, and adds significant new experiments and analysis. The work introduces a principled hierarchical scoring paradigm for instance-image goal navigation and leverages 3D Gaussian Splatting in a way not explored in prior work. And after this rebuttal period, we are sure to make our paper improve with the expanded conceptual insight, thorough ablations, and improved clarity.

---

### Meta-Review · Area_Chair_okEc · 2025-12-29

**Summary:**

This work proposes a framework for Instance Image-Goal Navigation that utilizes 3D Gaussian Splatting with a two-stage scoring mechanism. A global semantic scoring via CLIP and Mask-RCNN is followed by local geometric scoring using DINOv2. While the empirical results on HM3D are competitive, the issue highlighted by Reviewers `ZawG` and `nQaJ` is that the framework represents an engineering assembly of existing foundational models rather than a significant algorithmic advance. Furthermore, the core technical contribution like the local geometric scoring still relies on heuristic design choices and specific nonlinearities that lack strong theoretical grounding. Additionally, the requirement for a pre-requisite exploratory rollout limits the method's scope to map-based localization rather than robust online navigation. Therefore, the paper could be further improved so far based on the borderline scores.

**Reviewer Concerns:**

The authors provided responses regarding the choice of visual backbones and how the system handles incomplete maps, adding new data to support their claims. However, the disagreement about the method's value remained unresolved. Reviewer `nQaJ` argued that the mathematical rules for the local scoring section seemed based on trial-and-error, noting that the explanations for the specific formulas were still confusing. Reviewer `ZawG` remained unconvinced that the hierarchical approach was a significant invention, viewing it instead as a simple efficiency update, and questioned why standard existing segmentation tools would not work just as well. Additionally, Reviewer `YJ7j` pointed out that the requirement for a pre-built map is a serious limitation for robots that need to explore new environments in real-time.

**Reviewer Scores:**

The ratings were borderline, with Reviewers `YJ7j`, `Sks8`, and `ENRL` giving a borderline score because they acknowledge the performance on benchmarks and the system's speed. Reviewers `nQaJ` and `ZawG` maintained their lower scores throughout the discussion. These critics were the most concerned about the lack of new concepts, and since the rebuttal did not convince them that the method was more than just an engineering assembly.

---

### Decision · Program_Chairs · 2026-01-26

Reject